# Magnesium depletion by *Candida albicans* unleashes two unusual modes of colistin resistance in *Pseudomonas aeruginosa* with different fitness costs

Yu-Ying Phoebe Hsieh[1,2‡]*, Ian P. O'Keefe[3,4‡], Zeqi Wang[2], Wanting Sun[2,5], Hyojik Yang[3], Linda M. Vu[3], Nicole E. Smalley[6], Robert K. Ernst[3], Ajai A. Dandekar[6,7⊕], Harmit S. Malik[2,8⊕]

**1** Institute of Plant and Microbial Biology, Academia Sinica, Taipei, Taiwan, **2** Division of Basic Sciences, Fred Hutchinson Cancer Center, Seattle, Washington, United States of America, **3** Department of Microbial Pathogenesis, University of Maryland—Baltimore, Baltimore, Maryland, United States of America, **4** Department of Biochemistry and Molecular Biology, University of Maryland—Baltimore, Baltimore, Maryland, United States of America, **5** Department of Molecular Genetics & Microbiology, Duke University, Durham, North Carolina, United States of America, **6** Department of Medicine, University of Washington, Seattle, Washington, United States of America, **7** Department of Microbiology, University of Washington, Seattle, Washington, United States of America, **8** Howard Hughes Medical Institute, Fred Hutchinson Cancer Center, Seattle, Washington, United States of America

‡ These authors contributed equally as co-first authors to this work.
⊕ These authors contributed equally to this work.
* yhsieh@as.edu.tw

## Abstract

Increasing bacterial resistance to colistin, a vital last-resort antibiotic, is an urgent challenge. Previous studies have shown that $Mg^{2+}$ depletion enables *Pseudomonas aeruginosa* to become resistant to colistin. Here, we show that magnesium sequestration by *Candida albicans* also enables *P. aeruginosa* to evolve a nearly hundredfold higher level of colistin resistance through genetic changes in lipid A biosynthesis-modification pathways and a putative magnesium transporter. These mutations synergize with the $Mg^{2+}$-sensing PhoPQ two-component signaling system to remodel lipid A structures of the bacterial outer membrane in previously uncharacterized ways. One predominant mutational pathway involves early mutations in *htrB2*, a non-essential gene involved in lipid A biosynthesis, which enhances resistance but compromises outer membrane integrity, resulting in fitness costs and increased susceptibility to other antibiotics. A second pathway achieves increased colistin resistance independently of *htrB2* mutations without compromising membrane integrity. In both cases, reduced colistin binding to the bacterial membrane underlies resistance. Our findings reveal that $Mg^{2+}$ scarcity triggers novel evolutionary trajectories, leading to extremely high colistin resistance in *P. aeruginosa*.

**Data availability statement:** All relevant data are within the paper and its Supporting information files. All genome sequencing files are available from the NCBI Bioproject database (PRJNA1251133).

**Funding:** The following agencies funded this study: Cystic Fibrosis Foundation (https://www.cff.org/) HSIEH24F0 and HSIEH21F0-CI (to YPH) Cystic Fibrosis Foundation (https://www.cff.org/) ERNST23G0 and NIH (https://www.niaid.nih.gov/) AI104895 (to RKE) NIH (https://www.nigms.nih.gov/) R35 GM152107 (to AAD) Howard Hughes Medical Institute Investigator award (https://www.hhmi.org/) (to HSM). Funding agencies played no role in the study design or the decision to publish.

**Competing interests:** The authors have declared that no competing interests exist.

**Abbreviations:** BHI, brain heart infusion; CFU, colony-forming units; FLAT, Fast Lipid Analysis Technique; LPS, lipopolysaccharide; MALDI-TOF, matrix-assisted laser desorption/ionization time-of-flight; MICs, minimum inhibitory concentrations; MS, mass spectrometry; SEM, scanning electron microscopy; WT, wild type.

## Introduction

Antimicrobial resistance poses a significant global health challenge [1]. Prior research has focused on mechanisms that individual bacterial species use to evade antibiotics. However, microbial interactions can profoundly alter antibiotic resistance in ways that remain incompletely understood [2,3]. Complex microbial communities can protect susceptible species from antibiotics, modulate selection pressures [4,5], influence the emergence of resistant mutants [6], and alter the spectrum of resistance mutations [7]. Under these conditions, microbes must continuously adapt to complex selective pressures imposed both by antibiotics and their surrounding microbial partners, making it challenging to predict the evolutionary trajectory of antibiotic resistance. Since polymicrobial infections can exacerbate the spread of drug-resistant bacteria and pose a significant threat to human health and healthcare systems [8–10], understanding how microbial ecology shapes the evolution of antibiotic resistance is critical for developing effective therapies against polymicrobial infections.

*Pseudomonas aeruginosa* is a gram-negative bacterium commonly found in polymicrobial, drug-resistant infections [11,12]. Polymyxins, such as colistin (polymyxin E), are a last resort for treating multidrug-resistant *P. aeruginosa* infections [13,14]. In recent years, the number of colistin-resistant *P. aeruginosa* isolates has increased at an alarming rate [15]. Colistin targets gram-negative bacteria by electrostatically interacting with lipid A, the hydrophobic membrane anchor of lipopolysaccharide (LPS) in the outer leaflet of the outer bacterial membrane [16,17], eventually leading to membrane rupture and cell lysis [18]. LPS is the predominant constituent of the outer leaflet of the outer membrane that protects bacteria from environmental changes [19–21]. Bacteria can rapidly modify lipid A to facilitate adaptation to environmental perturbations [22,23]. For instance, upon exposure to colistin or $Mg^{2+}$ depletion, *P. aeruginosa* activates the PhoPQ and PmrAB two-component systems [24,25], which activate the expression of several lipid A-modifying proteins, including PagL and proteins encoded by the Arn operon. PagL alters the acyl chain number of lipid A by removing the acyl chain from the 3-position of lipid A [26]. Activation of Arn operons leads to the addition of 2-amino-2-hydroxy-L-arabinose (L-Ara4N) modification onto lipid A [27–29] (S1A and S1B Fig). These lipid A modifications reduce colistin binding to the outer membrane, thereby protecting against its antibacterial action [25,30,31]. Such lipid A modifications are conserved in gram-negative bacteria [31,32]. Yet, how polymicrobial environments alter conditions and mechanisms by which bacteria acquire colistin resistance remains underexplored.

*P. aeruginosa* coexists with the fungal pathogen *Candida albicans* in urinary tracts, chronic wounds, and the airways of people with cystic fibrosis [33–36]. In a previous study, we showed that *C. albicans* (and many fungi) sequester $Mg^{2+}$ from *P. aeruginosa* (and many gram-negative bacteria). $Mg^{2+}$ levels for such nutritional competition (<0.45 mM) are within the physiological range of $Mg^{2+}$ (0.1–0.8 mM) in several infection settings [37]. This $Mg^{2+}$ depletion not only instantaneously confers modest colistin resistance in *P. aeruginosa* cells but also alters their evolutionary trajectory, enabling them to acquire a nearly hundredfold higher level of colistin resistance [37]. We evolved WT *P. aeruginosa* (strain PAO1) to gain resistance in co-culture with

*C. albicans* (strain SC5314) by gradually increasing the colistin concentration from 1.5 to 192 µg/mL in brain heart infusion (BHI) media during daily transfers for 90 days (Fig 1A) [37]. We found that increased resistance to colistin always depended on either fungal co-culture or $Mg^{2+}$ depletion, as fungal removal or $Mg^{2+}$ supplementation reduced colistin resistance in all replicate populations [37]. In contrast, populations evolved without *C. albicans* did not show $Mg^{2+}$-dependent resistance and acquired canonical colistin resistance mutations, rarely observed in co-culture evolved populations [37].

Our findings suggested that low $Mg^{2+}$ conditions experienced by bacteria in polymicrobial environments [38] and during infection [39] can drive alternative modes of colistin resistance. However, the molecular nature of these mechanisms remained unclear. Here, we characterized the genetic, evolutionary, and biochemical mechanisms by which

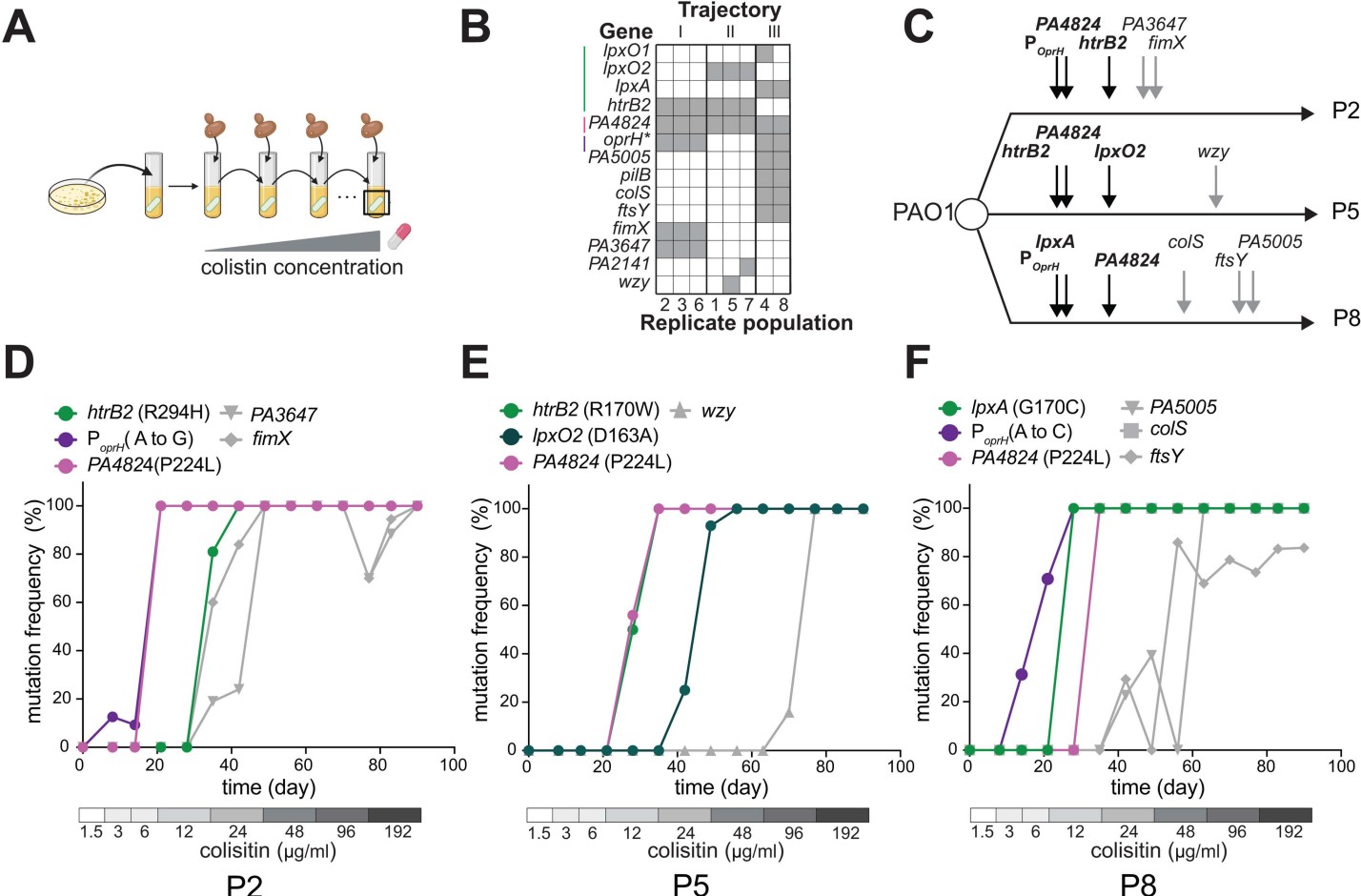

**Fig 1. Three distinct evolutionary trajectories lead to fungal-dependent colistin resistance under low $Mg^{2+}$ conditions. (A)** Schematic of evolving colistin-resistant *Pseudomonas aeruginosa* in co-culture with *Candida albicans*. Eight independent *P. aeruginosa* populations were passaged in BHI media with *C. albicans* in the presence of colistin. *C. albicans* cells were added at each passage, and the concentration of colistin was gradually increased from 1.5 to 192 µg/ml. **(B)** Eight co-culture-evolved populations exhibit three distinct trajectories of genetic mutation (shown as I, II, and III). Genes mutated in the eight populations are listed in rows, and mutations in a population are indicated as gray boxes (*indicates a non-coding promoter mutation). (A) and (B) summarize the experimental design and mutations identified in our prior study. **(C)** Summary of the order of mutation fixation in representative populations P2, P5, and P8. Mutations in lipid A biosynthesis genes, *PA4824*, and the promoter of the *oprH/phoP/phoQ* operon (labeled in black) were fixed in the early stage of evolution across three trajectories. **(D–F)** Temporal dynamics of mutation fixation in evolved populations P2 **(D)**, P5 **(E)**, and P8 **(F)** show that mutations in lipid A biosynthesis genes (*htrB2, lpxO2,* and *lpxA*), *PA4824,* and the *oprH/phoP/phoQ* promoter became fixed early during adaptation, when populations were exposed to 12–24 µg/ml colistin. The data underlying Fig 1D–1F can be found in S1 Data. Fig 1A is created with Biorender.com, Hsieh, P. (2026) https://BioRender.com/rewlecd.

PLOS Biology

these mutations confer novel modes of colistin resistance and investigated why they evolve exclusively under low Mg$^{2+}$ conditions. Our analyses reveal that colistin resistance arises via genetic epistasis between early-arising mutations that synergize with PhoPQ signaling, which is activated under low Mg$^{2+}$ conditions. Furthermore, we demonstrate that these mutations result in lipid A modifications distinct from those observed in the previously characterized colistin-resistant *P. aeruginosa*. These unique lipid A structures confer colistin resistance by reducing colistin binding to the membrane. Interestingly, many of these changes compromise bacterial membrane integrity, leaving colistin-resistant populations more susceptible to other antibiotics. Our study provides new molecular insights into how nutritional depletion of metal ions can affect bacterial antibiotic resistance.

## Results

### *P. aeruginosa* replicate populations acquire high levels of colistin resistance under low Mg$^{2+}$ conditions via genetic epistasis

We previously performed experimental evolution experiments, which revealed three trajectories leading to very high colistin resistance under low Mg$^{2+}$ (Fig 1A and 1B), all marked by convergent mutations in lipid A biosynthesis or modification genes (S1A Fig) and in the novel Mg$^{2+}$ transporter gene *PA4824*. We performed whole-genome sequencing of endpoint clones from all resulting colistin-resistant populations to comprehensively identify all mutations that could be responsible for higher colistin resistance. We found that the first trajectory (I) only acquired mutations in *htrB2*, *PA4824*, *fimX*, *PA3647*, and the promoter of the *oprH/phoP/phoQ* operon; the second trajectory (II) in *lpxO2*, *htrB2*, and *PA4824*; and the third trajectory (III) in *lpxA*, *PA4824*, PA5005, *pilB*, *colS*, *ftsY*, and the *oprH/phoP/phoQ* promoter.

We sought to understand how each of these trajectories acquired resistance over the course of the experimental evolution. For this purpose, we sequenced time-series samples from representative populations P2 (I), P5 (II), and P8 (III) (Fig 1B). Across all populations, mutations in lipid A genes (*htrB2*, *lpxO2*, *lpxA*), *PA4824*, and the *oprH/phoP/phoQ* promoter arose early and rapidly fixed as colistin concentrations increased (Fig 1C). In P2, the *oprH/phoP/phoQ* promoter and *PA4824* (P224L) mutations were fixed by day 21 (6 μg/ml), followed by *htrB2* (R294H) mutation by day 35 (12 μg/ml). In P5, *htrB2* (R170W) and *PA4824* (P224L) mutations were fixed by day 28 (12 μg/ml), and *lpxO2* (D163A) by day 42 (24 μg/ml). In P8, the *oprH/phoP/phoQ* promoter and *lpxA* (G170C) mutations were fixed by day 28 (12 μg/ml), followed by the *PA4824* (P224L) mutation by day 35. Our finding of convergent mutations in the same genes across independent populations strongly suggested that these mutations drive colistin resistance under low-Mg$^{2+}$ conditions.

To test their contributions to colistin resistance, we reconstructed single, double, and triple mutants—reflecting their early emergence in representative P2, P5, and P8 populations—in WT PAO1. We then measured MICs in *C. albicans*-spent BHI media, prepared from the supernatant of a fungal-saturated culture, to mimic a low Mg$^{2+}$ condition compared to standard BHI media (which has high Mg$^{2+}$) [37]. In each trajectory, combining the "early" mutations that were fixed by day 35 was sufficient to confer significantly higher colistin resistance than the WT strain (Fig 2A–2C). In all cases, triple-mutation-reconstructed strains were resistant to colistin concentrations equivalent to or higher than those used at the time points when these mutations were fixed. Consistently, these triple mutants showed significantly increased survival at colistin concentrations of 6–24 μg/ml in co-culture with *C. albicans* (S2 Fig). These results indicate that the MIC measurements in the low-Mg$^{2+}$ media are a reasonable proxy for increased resistance in our co-culture evolution experiments.

Among single mutations, *oprH/phoP/phoQ* promoter mutations and the *lpxA* mutation have the largest effects on resistance (Fig 2A and 2C). Intriguingly, although the same *PA4824* mutation (P224L) was found recurrently in all three representative populations, this mutation did not confer any degree of colistin resistance by itself. Further, positive epistasis was evident, especially in the P5 trajectory. Single mutations (*oprH/phoP/phoQ* promoter, *PA4824*, or *htrB2*) had no discernible effect on colistin resistance, whereas a double mutant (*htrB2+PA4824*) was modestly resistant, and the triple mutant (*htrB2+PA4,824+lpxO2*) was highly resistant (Fig 2B). The MIC of the P5-derived triple mutant (48 μg/ml) was 14-fold higher than would be expected from the addition of single mutants. In contrast, positive epistasis was far

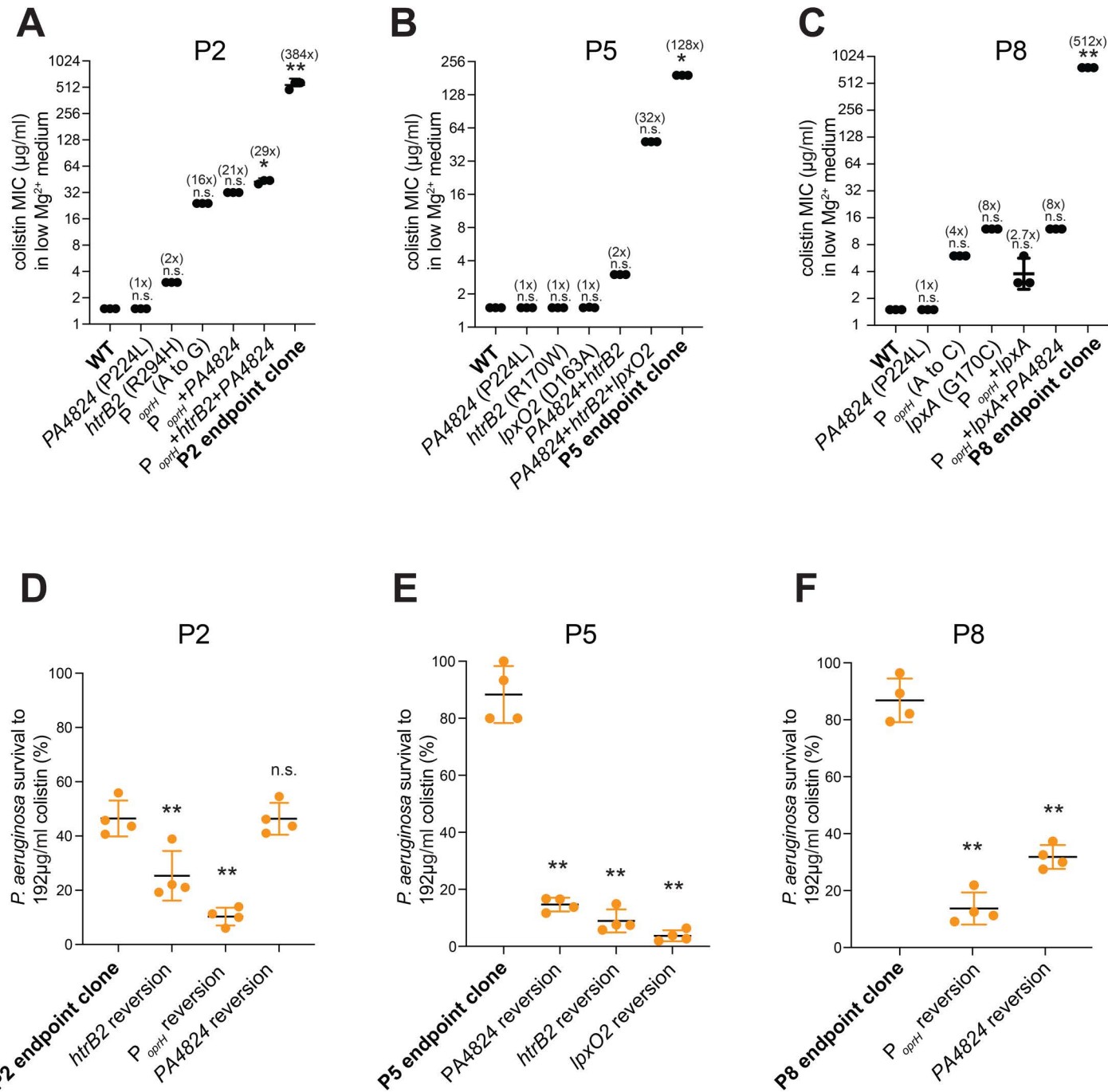

**Fig 2. Mutations in lipid A synthetic genes, *PA4824*, and the *oprH*/*phoP*/*phoQ* promoter are necessary and sufficient to explain colistin resistance. (A–C)** Early-onset mutations from representative populations P2 **(A)**, P5 **(B)**, and P8 **(C)** were reconstructed into the WT PAO1 strain. Strains with triple reconstructed mutations (but not single mutations) showed higher colistin MICs than WT PAO1 in low Mg$^{2+}$ media, with resistance levels higher than those they were selected during the experimental evolution. In all three representative populations, MICs of final evolved endpoint clones were higher than those of the early triple mutant reconstructed strains. Mean ± std of 3 biological replicates is shown (**$p < 0.01$, * $p < 0.05$, Kruskal–Wallis test with Dunn's multiple comparison correction). The MIC fold change relative to WT is indicated above the raw data. **(D–F) (D)** Reversion of *htrB2* (R294H) or *oprH*/*phoP*/*phoQ* promoter (A to G) mutation to the WT allele in a P2 endpoint clone reduced survival in 192 μg/ml colistin in the low Mg$^{2+}$ media. In contrast, reversion of PA4824 (P224L) had no significant effect. **(E)** Reversion of *PA4824* (P224L), *htrB2* (R170W), or *lpxO2* (D163A) mutations in a final P5 clone lowered survival to 192 μg/ml colistin in low Mg$^{2+}$ media. **(F)** Reversion of *PA4824* (P224L) or *oprH*/*phoP*/*phoQ* promoter (A to C) mutation in a final P8 clone lowered survival in 192 μg/ml colistin in low Mg$^{2+}$ media. Mean ± std of 4 biological replicates is shown (**$p < 0.01$, *$p < 0.05$, one-way ANOVA test with Dunn's multiple comparison correction). The data underlying Fig 2A–2F can be found in S2 Data.

less evident in the P2 and P8 trajectories (Fig 2A and 2C). For instance, in P8, a single oprH/phoP/phoQ promoter mutation or a lpxA mutation increased colistin MICs to 6 or 12 µg/ml, respectively, but combining them did not further elevate resistance. Together, these findings illustrate how the genetic background in which mutations arise and their interactions influence the evolutionary pathways to extremely high colistin resistance under low Mg$^{2+}$ conditions.

While early triple mutations explain the initial resistance, they were insufficient for the maximal resistance levels observed at the end of 90 days of experimental evolution, indicating that subsequent mutations can further contribute to enhanced resistance (Fig 2A–2C). Nevertheless, to test whether early-occurring mutations were necessary for the high degree of colistin resistance, we selected a single endpoint clone from each population that encoded all (shared) fixed mutations and the fewest unshared mutations. The exact mutations of each endpoint clone are listed in S1 Table. For each of the highly resistant endpoint clones from P2, P5, and P8, we reverted each early-occurring mutation to the WT allele. We then tested their survival when challenged with 192 µg/ml colistin in low Mg$^{2+}$ by measuring colony-forming units. We found that reverting mutations in htrB2, lpxO2, or the oprH/phoP/phoQ promoter significantly reduced bacterial survival. Reverting the htrB2 mutation to the WT allele in P2 and P5 reduced bacterial survival to ~20% or less (Fig 2D and 2E). Similarly, reverting the oprH/phoP/phoQ promoter mutation significantly reduced bacterial survival in P2 (20%) and P8 (10%) (Fig 2D and 2F), while reverting the lpxO2 mutation reduced P5 survival to 3% (Fig 2E). In contrast, reverting the PA4824 mutation significantly lowered bacterial survival in P5 and P8, but not in P2 (Fig 2D–2F), suggesting that other mutations might compensate for PA4824 evolution in P2. Similarly, we found these reversion strains had lower colistin MIC in the low-Mg$^{2+}$ media (S3 Fig), following a similar trend as the colistin survival assay. Our findings establish the cadence and causality of the genetic mutations observed in three distinct evolutionary trajectories. They reveal that "early" mutations in htrB2, lpxO2, oprH/phoP/phoQ promoter, and PA4824 were not only sufficient to evolve initial moderate colistin resistance but were also necessary to later acquire much higher colistin resistance under low Mg$^{2+}$ conditions.

## Novel, distinct lipid A structures underlie low Mg$^{2+}$-dependent colistin resistance

Our experimental evolution uncovered mutations involved in lipid A biosynthesis (Figs 1B and S1A) that confer high levels of colistin resistance. For example, htrB2 is mutated in two of three evolutionary trajectories, while other lipid A biosynthesis genes (lpxO1, lpxO2, and lpxA) are mutated in at least one trajectory (Fig 1B). To determine their impact on lipid A, we used the fast lipid analysis technique (FLAT) paired with matrix-assisted laser desorption/ionization time-of-flight (MALDI-TOF) mass spectrometry (MS) [40] to characterize the lipid A structures of all replicate populations and compared them to WT PAO1.

WT PAO1 grown in BHI (high Mg$^{2+}$) synthesized expected lipid A structures, with penta-acylated lipid A ($m/z = 1445.86$), which varies in 2-hydroxylation status ($m/z = 1461.85$) and phosphorylation status ($m/z = 1365.89, 1525.82$) (Figs 3A and S4A) [26,41]. In contrast, PAO1 grown in fungal-spent BHI (low Mg$^{2+}$) synthesized lipid A that was hexa-acylated, with PagP-mediated palmitate addition ($m/z = 1,684, 1,700$) [42] or a single L-Ara4N modification ($m/z = 1,576, 1,592$) [43] (Figs 3B and S4B), which is mediated by the PhoPQ and PmrAB two-component systems [24,25]. In contrast to WT PAO1, all eight colistin-resistant populations (P1 through P8) showed mass spectra unique to each evolutionary trajectory and distinct from WT in both high and low Mg$^{2+}$ conditions (S5 Fig), suggesting that they were genetically determined and not induced by low Mg$^{2+}$.

Our analysis identified several lipid A structures in each of the three evolutionary trajectories that have not been previously reported in P. aeruginosa, including those in colistin-resistant strains [44,45]. These distinct lipid A structures include species with various acylation and L-Ara4N modifications. For example, populations P2, P3, and P6 (trajectory I) synthesized penta-acylated lipid A with PagP-mediated palmitate addition but lacked HtrB2-mediated laurate addition ($m/z = 1632.96$) (Figs 3C and S4C, S2 Table). In contrast, populations P1, P5, and P7 (trajectory II) exhibited mixtures of two distinct lipid A structures without PagP-mediated palmitate addition. Instead, these trajectory II variants exhibited either tetra-acylated lipid A that lacked HtrB2-mediated acylation and LpxO2-mediated hydroxylation ($m/z = 1378.75$),

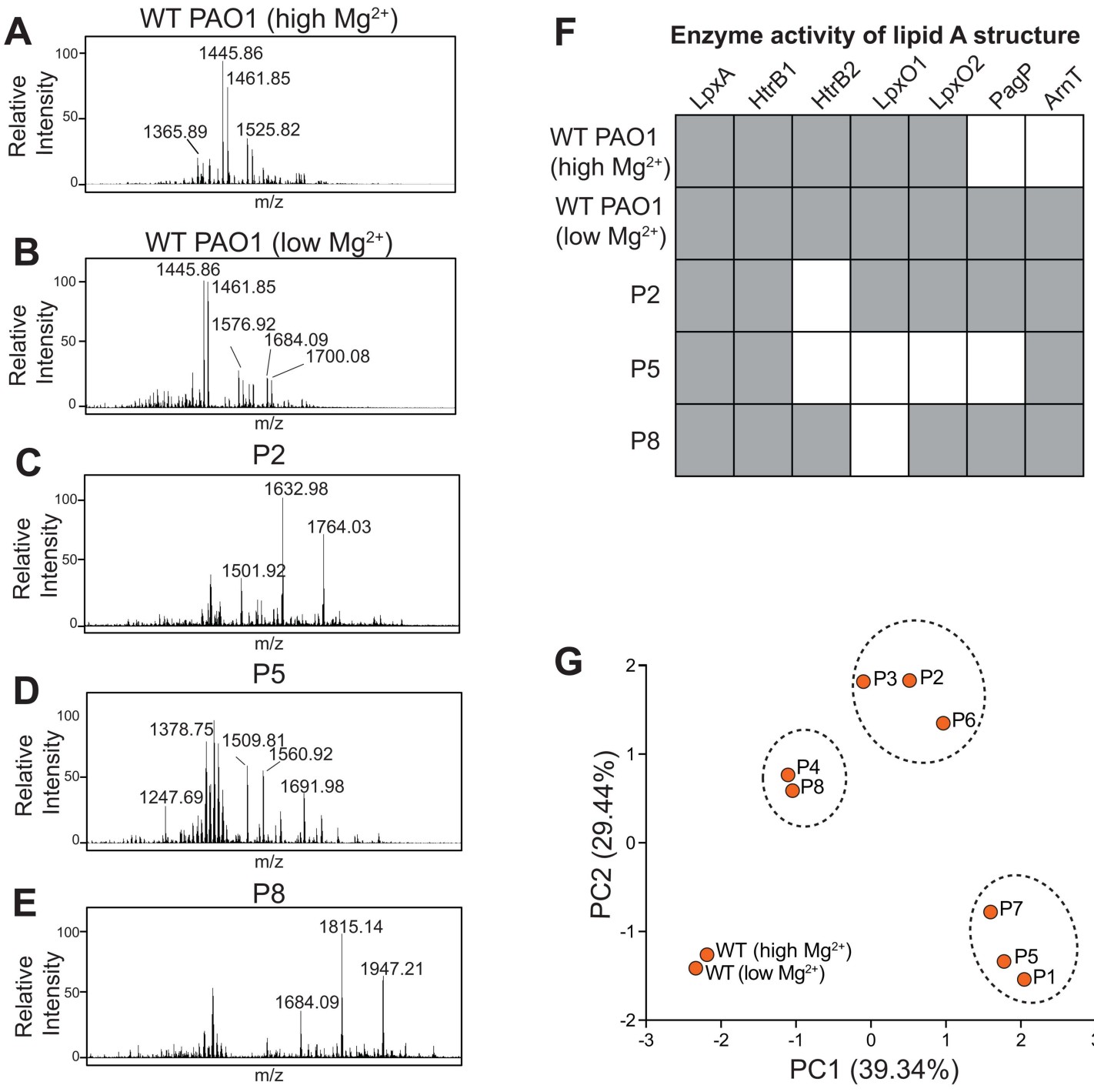

**Fig 3. Novel, distinct lipid A structures in each of three evolutionary trajectories of low Mg$^{2+}$-dependent colistin-resistant populations. (A)** Mass spectra of WT PAO1 in high Mg$^{2+}$ media reveal major ions at m/z values 1365.89, 1445.86, 1461.85, 1525.82, and 1451.82. This lipid A is penta-acylated, with both single and double 2′-hydroxylation mediated by LpxO1/LpxO2, indicated by differences of approximately 16 *m/z*, and single, double, and triple phosphorylation status, indicated by differences in approximately 80 *m/z* from the base peak. **(B)** Mass spectra of WT PAO1 in low Mg$^{2+}$ media reveal additional hexa-acylated lipid A with PagP-mediated palmitoylation, as seen by ions 1684.09 and 1700.08, or single L-Ara4N addition from the base penta-acylated structure, shown by the ion at *m/z* = 1576.92. **(C)** Mass spectra analyses of P2 cells reveal penta-acylated lipid A with PagP-mediated palmitate addition but lacking HtrB2-mediated laurate addition, leading to *m/z* values of 1501.01, 1632.98, and 1764.03. **(D)** Mass spectra analyses of P5 cells reveal lipid A either lacking HtrB2-mediated acylation and LpxO2-mediated 2-hydroxylation (*m/z* = 1378.75) or containing HtrB2-mediated acylation

but lacking both LpxO1 and LpxO2-mediated 2-hydroxylation ($m/z = 1560.92$). **(E)** Mass spectra analyses of P8 reveal hexa-acylated lipid A with PagP-mediated palmitoylation but lacking LpxO1-mediated hydroxylation ($m/z = 1684.09$, $1815.14$, $1947.21$). In addition, lipid A in P2, P5, and P8 cells exhibits single or double L-Ara-4-N addition, shown by differences of approximately 131 $m/z$. **(F)** A summary table of the activity of lipid A biosynthesis and modification enzymes. Lipid A structures obtained from MALDI-TOF MS were used to infer the activity of each enzyme (gray box indicates functional enzyme). The WT strain and endpoint clones (P2, P5, and P8) were analyzed in high and low Mg2+ conditions. In the endpoint clones, the activity of these enzymes was similar in both conditions. **(G)** PCA of lipid A mass spectra from all eight evolved populations and the laboratory-adapted WT strain PAO1 based on the presence or absence of lipid A peaks. PC1 and PC2 are shown, with 39.34% and 29.44% of the variance explained, respectively. The denoted groups cluster together based on observed lipid A structures from FLAT. The data underlying Fig 3G can be found in S3 Data.

or penta-acylated lipid A with HtrB2-mediated laurate addition, but without LpxO1 and LpxO2-mediated hydroxylation ($m/z = 1560.92$) (Figs 3D and S4D, S2 Table). Finally, populations P4 and P8 (trajectory III) had hexa-acylated lipid A containing PagP-mediated palmitate addition but lacking LpxO1-mediated hydroxylation ($m/z = 1815.14$) (Figs 3E and S4E, S2 Table). The lipid A of these evolved populations had zero, single, or double L-Ara4N additions, shown by differences of approximately 131 $m/z$ between ions in Fig 3C–3E. These results demonstrate that selection for colistin resistance in low Mg2+ leads to dramatic diversification of lipid A structures (Figs 3F and S4). Principal component analysis (PCA) of the MS data revealed that lipid A structures of the eight evolved populations formed three separate clusters (Fig 3G), each distinct from WT PAO1, mirroring the three evolutionary trajectories we had observed previously (Fig 1B).

To identify the genetic mutations underlying these biochemical changes, we analyzed early-onset triple mutants associated with moderate colistin resistance (described above). We found that early-onset mutations in P2 and P5 can fully recapitulate the lipid A structures of endpoint clones (S6A and S6B Fig). In contrast, triple mutants of P8 partially recapitulate the final lipid A structures but still contain LpxO1-mediated 2′-hydroxylation (S6C Fig). PCA analysis also showed that the lipid A MS spectra of the early-onset triple mutants were closer to those of the corresponding endpoint strains (S6D Fig). Since these early-occurring triple mutants showed increased resistance compared to the ancestral strain (Fig 2A–2C), we conclude that the unique lipid A changes associated with these early mutations causally contributed to colistin resistance. Further, this comparison suggests that other late-occurring mutations in these three lineages have minimal effects on lipid A structures, indicating they might further enhance resistance via a lipid A-independent mechanism.

We also assessed the necessity of specific mutations for the distinct lipid A structures found in each trajectory by performing FLAT on each mutant-reversion strain. Reverting individual mutations in *htrB2* to the wild type (WT) restored the laurate moiety (S7A and S7D Fig), while restoring *lpxO2* restored 2′-hydroxylation of lipid A (S7F). These results suggested that both *htrB2* and *lpxO2* mutations disrupted enzyme function. To further test this possibility, we engineered complete deletions of either gene in the endpoint isolates. We found that *lpxO2* deletion in the P5 endpoint strain did not alter its MIC, confirming that the *lpxO2* mutation we found during experimental evolution is effectively a null allele (S3B Fig). However, *htrB2* deletion in P2 and P5 endpoint clones resulted in further reductions in MICs, indicating that the evolved variants are partial loss-of-function alleles (S3A and S3B Fig). We concluded that, despite their importance to lipid A biosynthesis in *P. aeruginosa* [46,47], loss-of-function mutations in *lpxO2* and *htrB2* can nevertheless increase colistin resistance under low Mg2+ conditions by altering lipid A structures in novel ways. Consistent with this idea, reversion of the individual mutations in the endpoint clones results in WT-like lipid A structures (S7I Fig). Finally, reversion of *oprH/phoP/phoQ* promoter mutations prevented the addition of PagP-mediated palmitoylation (S7C and S7G Fig), a modification activated by PhoPQ. This result suggests that mutations in the *oprH/phoP/phoQ* promoter are likely to affect downstream effectors triggered by the PhoPQ pathway.

In contrast to *htrB2*, *lpxO2*, and the *oprH/phoP/phoQ* promoter, mutations in *PA4824* and *lpxA* appear to contribute to colistin resistance without altering lipid A. Reversion of the *PA4824* mutation (P224L) in all three endpoint clones had no discernible effect on lipid A structures (S7B, S7E, and S7H Fig). *lpxA* encodes the first, essential enzyme in lipid A biosynthesis (S1 Fig) [48]. Despite the same *lpxA* mutation (G170C) being found in the trajectory III, P8 endpoint clones still had LpxA-mediated acylation in their lipid A structures, indicating this mutation does not impair the enzymatic function of LpxA.

Overall, our findings revealed that high colistin resistance in low $Mg^{2+}$ conditions depends on unique lipid A structures that arise via mutations in *htrB2* and *lpxO2*, or promoter mutations in the *oprH/phoP/phoQ* operon. In contrast, mutations in *PA4824* and *lpxA* drive higher colistin resistance without apparently altering lipid A structures.

## The PhoPQ pathway synergizes with early-arising mutations to potentiate colistin resistance under low $Mg^{2+}$ conditions

One of the most interesting aspects of our experimental evolution studies was that the potentially colistin-resistant mutations observed in low $Mg^{2+}$ conditions were not observed in high $Mg^{2+}$. We investigated why this might be the case. Consistent with many prior studies [28,29], we found that bacteria activate both the PhoPQ and PmrAB two-component systems to modify their membranes in low $Mg^{2+}$ (S8 Fig). However, our identification of *oprH/phoP/phoQ* promoter mutations in two evolutionary trajectories (represented by P2 and P8), compared to none in the *pmrAB* operon (Fig 1B), suggests that these two systems played distinct roles in the development of colistin resistance during experimental evolution in low $Mg^{2+}$. To test this possibility, we deleted the genes encoding the transcriptional factors PhoP and PmrA, individually or in combination. We first tested the effects of these mutations in the early-onset triple-mutant backgrounds of P2, P5, and P8. The *phoP* deletion significantly reduced the colistin MIC to 3 µg/ml, whereas the *pmrA* mutation had only a mild effect. Moreover, *phoP pmrA* double deletion mutants showed no further reduction in resistance (S9A Fig). This result suggests that PhoPQ, but not PmrAB, plays a dominant role in driving colistin resistance at least in the initial stages of acquiring increased colistin resistance under low $Mg^{2+}$ conditions. Our analysis of endpoint clones from the P2, P5, and P8 populations corroborated these findings (S9B Fig), revealing that even high colistin resistance in endpoint clones was much more dependent on PhoPQ than PmrAB.

Our experiments already indicated that mutations in the *oprH/phoP/phoQ* promoter in P2 and P8 endpoint clones affected *phoP* levels. However, it remained possible that promoter mutations conferred colistin resistance by affecting both *oprH* and *phoP* expression. To test this possibility, we deleted either *oprH* or *phoP* from the P2 and P8 endpoint clones. In both cases, we found that deleting *phoP*, but not *oprH*, significantly decreased the colistin MIC in the low-$Mg^{2+}$ media (S10 Fig). These results suggested that the promoter mutations were selected for their ability to activate PhoPQ, not OprH, to further increase colistin resistance.

To examine how these promoter mutations affect PhoPQ-regulated genes, we introduced each mutation into the WT strain and measured the expression of two known PhoPQ targets: *arnC* [49] and *pagP* [50]. In both high and low $Mg^{2+}$ media, we found that *arnC* and *pagP* were upregulated in the promoter-mutation-reconstructed strains relative to WT (Fig 4A and 4B). Using lipid A structural analysis, we also found that both *phoPQ* promoter mutations introduced into WT PAO1 were sufficient to activate PhoPQ-mediated lipid A modifications typically only observed in low $Mg^{2+}$ conditions, even in high $Mg^{2+}$, recapitulating the same L-Ara4N modification and PagP-mediated acylation observed in populations P2 and P8 (Fig 4C–4E). In contrast, loss of *phoP* in P2 and P8 endpoint clones completely abrogated L-Ara4N modification and PagP-mediated acylation of LPS (S11 Fig), consistent with previous reports [24,30]. These results confirm that either promoter mutation is sufficient to drive lipid A modifications that enhance colistin resistance. Collectively, our findings reveal that PhoPQ activation potentiated entirely novel evolutionary trajectories of colistin resistance, which were favored under low $Mg^{2+}$ conditions.

## Two of three evolved lineages acquired colistin resistance by reducing colistin binding, but have compromised membrane integrity

We next investigated the potential consequences of the unique lipid A modifications in colistin-resistant cells on cellular structures. Using scanning electron microscopy (SEM), we found that endpoint clones from all three evolutionary lineages exhibited abnormal morphology compared to WT cells. These abnormalities were more severe in the P2 and P5 endpoint clones, which exhibited deformed membranes in both high and low $Mg^{2+}$ conditions, with P5 cells being elongated in the

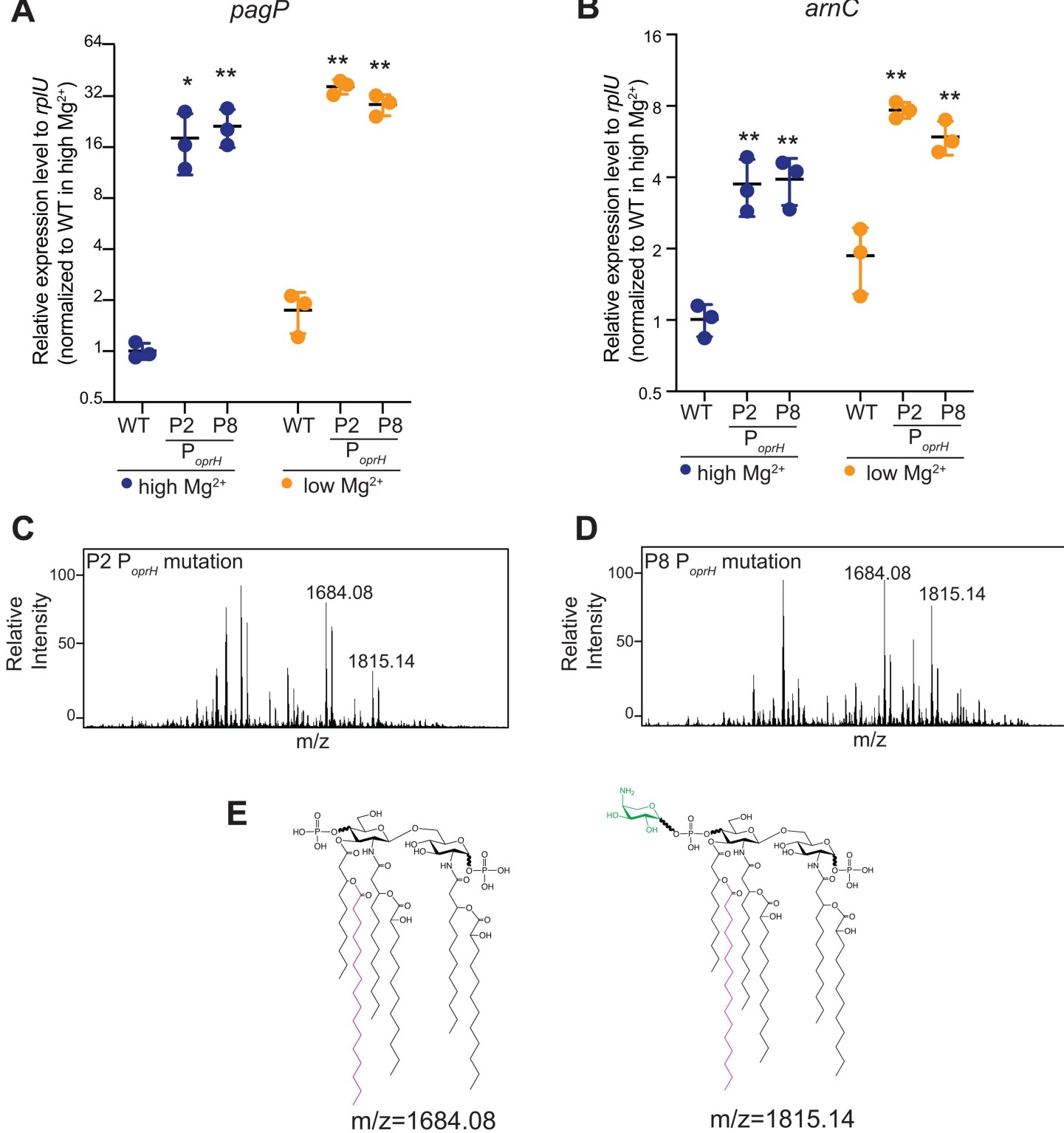

**Fig 4. The PhoPQ pathway synergizes with early-arising mutations to confer colistin resistance. (A, B)** Mutations in the promoter of the *oprH*/*phoP*/*phoQ* operon in replicate populations P2 and P8 cause the activation of PhoPQ-regulated genes *pagP* (A) and *arnC* (B) in both high and low $Mg^{2+}$ conditions (* $p < 0.05$, ** $p < 0.01$, one-way ANOVA test with Dunn's multiple comparison correction). **(C, D)** Mass spectra of lipid A modifications in

PAO1 containing the promoter mutation in the *oprH*/*phoP*/*phoQ* operon from either P2 (C) or P8 (D) confirm the PhoPQ activation, with the $m/z$ peak representing PhoPQ activity labeled in bold, where 1684.08 represents lipid A with PagP-mediated acylation but without L-Ara4N addition, and 1815.14 represents lipid A with PagP-mediated acylation and L-Ara4N addition. **(E)** Molecular structures of lipid A at $m/z = 1684.08$, 1815.14 with PagP-mediated acylation (in magenta) and L-Ara4N addition (in green). The data underlying Fig 4A and 4B can be found in S4 Data.

low Mg$^{2+}$ condition. In contrast, P8 endpoint clones exhibited less severe abnormalities, with slightly shorter cells than WT in both conditions (Figs 5A and S12). Our findings suggested that cell morphology was affected in all three trajectories of cells that acquired high colistin resistance in low Mg$^{2+}$, but to variable degrees.

Inspired by the SEM analyses, we investigated whether mutations that led to increased colistin resistance might have compromised the integrity of the outer bacterial membrane. In a first assay, we grew endpoint clones in LB media and found that P2 and P8 endpoint clones exhibited similar viability to WT in LB alone, whereas P5 cells were growth-impaired even in LB alone. We then exposed the endpoint clones to membrane-perturbing agents such as SDS and EDTA [51] in LB media. Upon treatment with these agents, we found that P2 and P5 endpoint clones were significantly more sensitive than WT cells, while P8 cells remained relatively unaffected (Fig 5B). In a second assay, we assessed outer membrane permeability in both high and low Mg$^{2+}$ media using the fluorometric probe NPN, which fluoresces upon entering the periplasm and binding phospholipids when membrane integrity is compromised. In high Mg$^{2+}$, we found that P2 and P5 endpoint clones exhibited increased NPN uptake, whereas P8 did not (Fig 5C). However, in low Mg$^{2+}$ media, all three endpoint clones demonstrated higher NPN uptake than WT. Based on these findings, we conclude that all three endpoint clones have compromised membrane integrity relative to the WT strain, but P2 and P5 cells have more severe membrane defects than P8 cells.

Although P2 and P5 endpoint clones have more permeable membranes, they exhibited greater resistance to polymyxin antibiotics, including colistin (polymyxin E) (Fig 5D) and polymyxin B (S13A Fig) than WT cells. To investigate how membrane-compromised cells acquire increased resistance to outer membrane-targeting antibiotics, we quantified polymyxin B binding to *P. aeruginosa*, using polymyxin B binding as a surrogate for all polymyxin antibiotics, including colistin. For this, we used dansyl-labeled polymyxin B [51], which fluoresces upon binding the hydrophobic portion of bacterial membranes. We validated the assay using Δ*phoP* Δ*pmrA* mutant cells and WT cells, finding that mutant cells demonstrate increased binding to polymyxin B (S13B Fig). Next, we evaluated polymyxin B binding to P2, P5, and P8 endpoint cells. In low Mg$^{2+}$ media, we found that P2 and P5 had a 20%–30% reduction in polymyxin B binding compared to WT, while P8, the most colistin-resistant clone, had the lowest levels of polymyxin B binding (Fig 5E). Similar trends were observed in the high Mg$^{2+}$ media (Fig 5E). Our findings show that P2, P5, and P8 endpoint clones acquire a high degree of colistin resistance by reducing colistin binding. Two of these lineages do so by altering lipid A, compromising their membrane integrity—a novel mechanism of colistin resistance.

## Two modes of high colistin resistance lead to distinct patterns of fitness tradeoffs

To further investigate why the colistin-resistance mutations we have characterized in this study were not observed in high Mg$^{2+}$ conditions, we compared the relative fitness of all three endpoint clones to that of WT PAO1 in high or low Mg$^{2+}$ media, in the absence of colistin. We found that the severity of their membrane defects correlated with fitness costs for the three evolved populations. P2 and P5 endpoint clones had significantly lower fitness than WT in both conditions, whereas P8 endpoint clones did not (Fig 6A). Even the early triple-mutant reconstructed strains of P2 and P5 exhibited a significant fitness reduction compared to WT (S14 Fig), suggesting that a tradeoff between increased colistin resistance and bacterial membrane integrity manifests during early evolution. Despite exhibiting fitness costs under high Mg$^{2+}$, triple mutants and endpoint clones of P2 and P5 under high Mg$^{2+}$ had much lower colistin MICs than under low Mg$^{2+}$ (S15A–S15C Fig). These MICs were less than the colistin concentration we applied during experimental evolution, indicating these mutations could not have been selected under high Mg$^{2+}$ conditions.

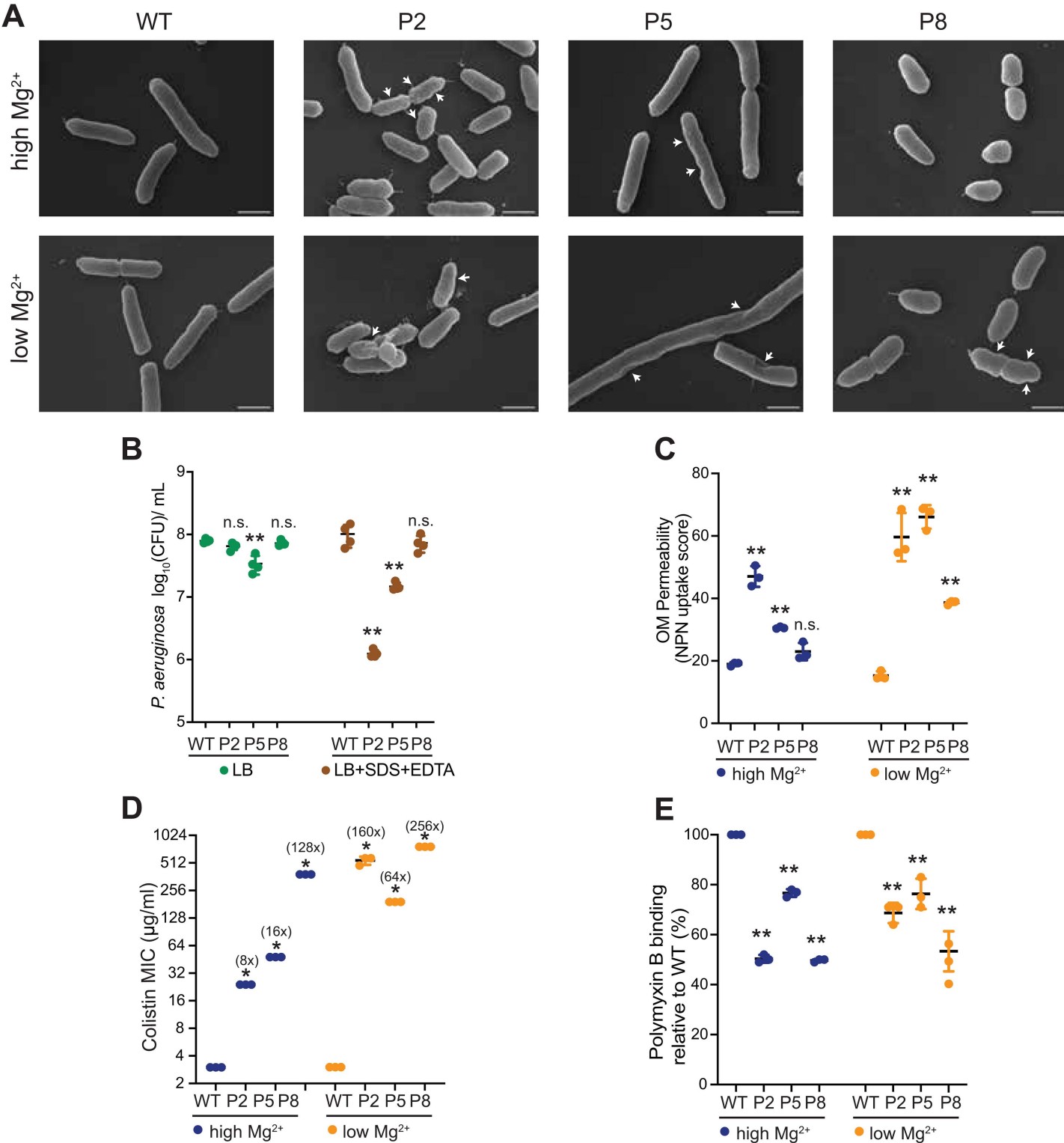

**Fig 5. Replicate populations P2 and P5, but not P8, have membrane defects. (A)** Scanning electron microscopic images of P2, P5, and P8 endpoint strains in high and low Mg$^{2+}$ media. In high Mg$^{2+}$, P2 and P5, but not P8, have discernible dents or kinks in the cell membrane (white arrows). In low Mg$^{2+}$, P5 showed severe membrane deformation. All three displayed altered cell shapes compared to WT PAO1. The scale bar indicates 1 μm. **(B)** Log-phase cells were serially diluted on an LB plate (green) or an LB plate supplemented with SDS and EDTA (brown) to assay bacterial resistance to

membrane-perturbing agents. Cells from the P2 and P5, but not P8 lineages, were more sensitive to membrane stress. Mean ± std of 4 biological replicates is shown (**$p < 0.01$, one-way ANOVA test with Dunn's multiple comparison correction). **(C)** An NPN assay was used to measure outer membrane permeability of WT, P2, P5, and P8. All three lineages had increased outer membrane permeability in the low-Mg$^{2+}$ media. Mean ± std of 3 biological replicates is shown (**$p < 0.01$, one-way ANOVA test with Dunn's multiple comparison correction). **(D)** Colistin MIC of endpoint clones in high and low Mg$^{2+}$ media. The MIC fold change relative to WT is indicated above the raw data. Mean ± std of 3 biological replicates is shown (* $p < 0.05$, Kruskal–Wallis test with Dunn's multiple comparison correction). **(E)** Dansyl-polymyxin B was used to quantify polymyxin B binding to WT PAO1 and endpoint clones in high and low-Mg$^{2+}$ media. All three endpoint clones had less binding to polymyxin B than WT in both conditions. Mean ± std of 3 biological replicates is shown (**$p < 0.01$, one-way ANOVA test with Dunn's multiple comparison correction). The data underlying Fig 5B–5E can be found in S5 Data.

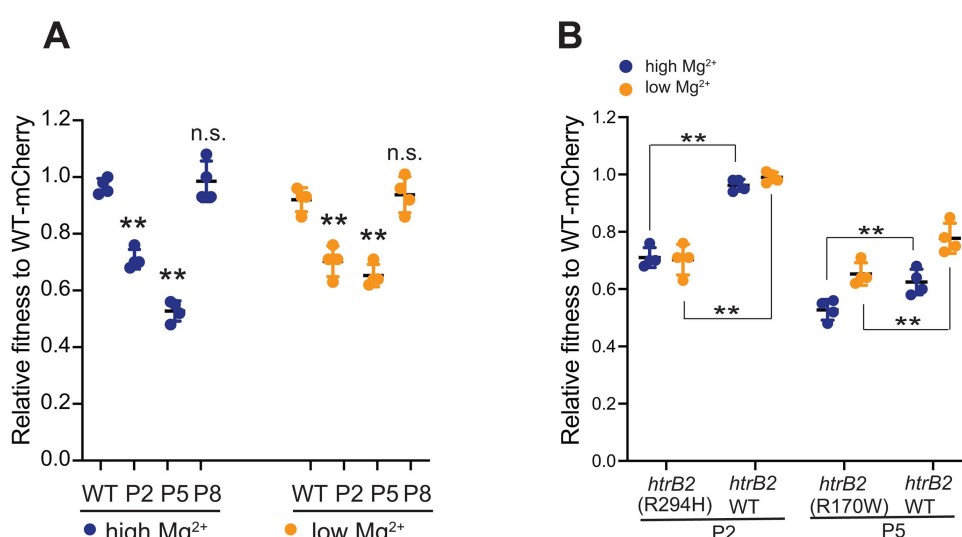

**Fig 6. Colistin-resistant endpoint clones of P2 and P5 show fitness costs due to *hrtB2* mutations. (A)** P2 and P5, but not P8, showed reduced fitness in high and low Mg$^{2+}$. A competitive fitness assay was used to measure the fitness of P2, P5, and P8, relative to WT PAO1. Mean ± std of 3 biological replicates is shown (**$p < 0.01$, one-way ANOVA test with Dunn's multiple comparison correction). **(B)** *htrB2* reversion increases the fitness of P2 and P5 in high and low Mg$^{2+}$ conditions. A competitive fitness assay was used to assess the fitness of each revertant strain relative to WT in high Mg$^{2+}$ (blue) and low Mg$^{2+}$ (orange) conditions without colistin. The mean ± std of 4 biological replicates is shown (**$p < 0.01$, *$p < 0.05$, one-tailed Mann–Whitney *U* test). The data underlying Fig 6A and 6B can be found in S6 Data.

Based on mutations common to P2 and P5 but not found in P8 (Fig 1B), we hypothesized that the loss-of-function and missense mutations in *htrB2*, an enzyme involved in lipid A acylation [47], were primarily responsible for membrane defects in P2 and P5. We tested this hypothesis by reverting the *htrB2* mutation to the WT allele in P2 and P5 endpoint clones. In both cases, we found that the *htrB2* reversion significantly improved bacterial fitness in low and high Mg$^{2+}$ media (Fig 6B). Additionally, an Δ*htrB2* mutant in WT PAO1 was itself sufficient to lower membrane integrity [47], further confirming that *htrB2* mutations enhance colistin resistance at the expense of membrane integrity and bacterial fitness.

In contrast to P2 and P5, the P8 lineage followed a distinct evolutionary path by acquiring a mutation in *lpxA,* the first essential gene in the biosynthesis of lipid A. P8 endpoint clones showed only minimal impairment in bacterial membrane integrity and exhibited no fitness costs compared to WT cells, suggesting that this path incurs fewer tradeoffs. However, we found that the early P8 triple mutant also had significantly reduced fitness compared to WT in high, but not low, Mg$^{2+}$ media (S14 Fig). This initial fitness cost may have been sufficient to prevent this trajectory from emerging under high Mg$^{2+}$, whereas low Mg$^{2+}$-dependent PhoPQ activation might have offset early fitness defects in P8. However, this fitness cost in high Mg$^{2+}$ conditions is ameliorated in P8 endpoint clones (Fig 6A), suggesting that other late-occurring mutations (*PA5005*, *colS*, and *ftsY*) could suppress the previously acquired fitness defect.

## Dual modes of colistin resistance in low Mg$^{2+}$ lead to distinct susceptibility to other antibiotics

Our analyses show that membrane integrity defects lower colistin binding, thereby conferring colistin resistance. However, we hypothesized that membrane integrity defects might increase susceptibility to other antibiotics that target intracellular processes by lowering the barrier to their penetration [52]. To test this possibility, we examined the susceptibility of the P2, P5, and P8 end-point clones under high- and low-Mg$^{2+}$ conditions to three antibiotics that are typically ineffective against *P. aeruginosa*: vancomycin, which targets the cell wall [53]; rifampicin, which targets RNA polymerase [54]; and azithromycin, which inhibits protein synthesis [55] (Fig 7A).

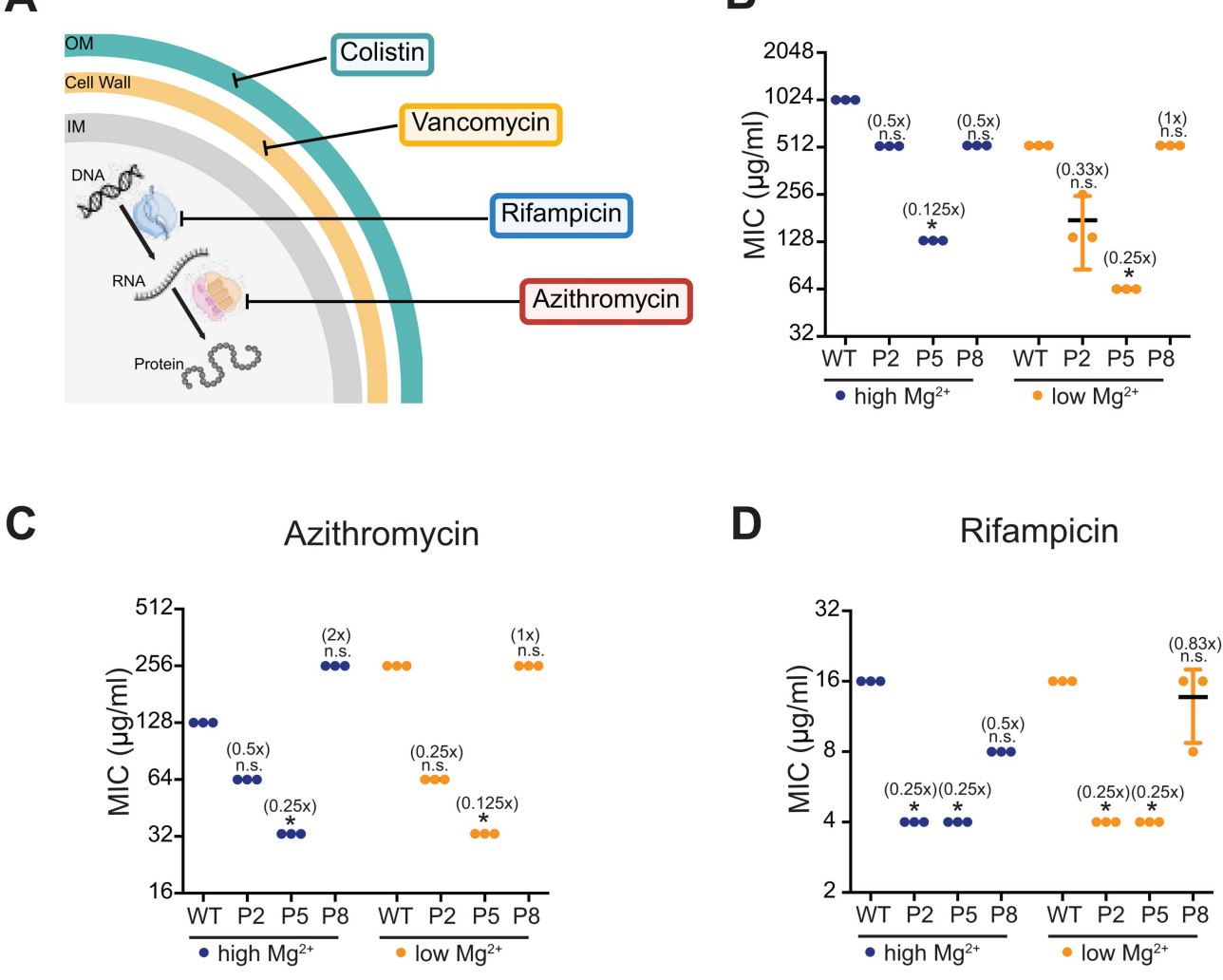

**Fig 7. Colistin-resistant endpoint clones of P2 and P5 are more susceptible to other antibiotics. (A)** A schematic of the mode of action of different antibiotics and where they act in the bacteria. OM, outer membrane; IM, inner membrane. **(B–D)** The resistance to three distinct antibiotics in high and low Mg$^{2+}$ was measured using standard MIC assays with three independent clones of WT, P2, P5, and P8 endpoint strains. MICs to vancomycin **(B)**, azithromycin **(C)**, and rifampicin **(D)** are shown. Mean±std of 3 biological replicates is shown (* $p < 0.05$, Kruskal–Wallis test with Dunn's multiple comparison correction). The fold change in MIC relative to WT is shown above the raw data. The data underlying Fig 7B–7D can be found in S7 Data. Fig 7A is created with Biorender.com, Hsieh, P. (2026) https://BioRender.com/ddivp31.

The P2, P5, and P8 end-point clones exhibited much higher colistin resistance than WT in both low- and high-Mg$^{2+}$ media (Fig 5D). P2 and P5 endpoint clones, which had *htrB2* mutations, more severe membrane defects, and increased fitness costs, exhibited significantly lower minimum inhibitory concentrations (MICs) than WT cells for all three antibiotics (Fig 7B–7D). In contrast, P8 endpoint clones, which harbor WT *htrB2* and exhibit only mild membrane defects and fitness costs, showed minimal increases in susceptibility to other antibiotics (Fig 7B–7D). This correlation (S16 Fig) supports our inference that increased susceptibility to other antibiotics in colistin-resistant strains stems from increased membrane permeability.

Importantly, such evolutionary trade-offs were observed only in low Mg$^{2+}$-dependent resistant strains. In contrast, *P. aeruginosa* strains that evolved colistin resistance in high Mg$^{2+}$ through previously characterized resistance mutations [37] did not exhibit comparable changes in permeability, antibiotic susceptibility, or fitness (S17 Fig). Thus, our studies reveal two evolutionary paths of colistin resistance in *P. aeruginosa*, both uniquely enabled by low Mg$^{2+}$, with distinct genetic and biochemical underpinnings and divergent fitness tradeoffs (Fig 8).

## Discussion

Environmental conditions and microbial interactions with cohabiting species can strongly influence the evolution of microbial traits, such as antibiotic resistance [56–58]. Our study investigated how fungal-driven Mg$^{2+}$ depletion enables evolutionary pathways in *P. aeruginosa* that enable cells to acquire extremely high colistin resistance [37]. We identified convergent mutations in lipid A biosynthesis and modification genes, the *phoPQ* operon promoter, and *PA4824* (a putative Mg$^{2+}$ transporter) that caused lipid A modifications and recapitulated early steps toward resistance under Mg$^{2+}$-depleted conditions. This increased colistin resistance relied on synergistic interactions between the early mutations and PhoPQ activation under low Mg$^{2+}$ conditions. However, the same mutations incur high fitness costs and don't confer resistance under high Mg$^{2+}$ conditions, explaining why they are unlikely to occur in Mg$^{2+}$-replete conditions.

Our analyses unexpectedly revealed two unique genetic and biochemical pathways to colistin resistance in low Mg$^{2+}$ conditions. The first pathway (trajectories I and II, *e.g.,* P2 and P5) relies on partial loss-of-function mutations in *htrB2* (Fig 3E and 3F)*,* which typically adds a laurate moiety to lipid A in the outer bacterial membrane of WT *P. aeruginosa* cells [47]. As a result, populations with *htrB2* mutations have compromised outer membrane integrity, reducing colistin binding to the membrane. These genetic changes confer substantial fitness costs and heighten susceptibility to other antibiotics targeting intracellular processes. This finding represents a unique mode of the phenomenon of collateral sensitivity, the previously described phenomenon in which increased resistance to one antibiotic leads to increased susceptibility to another [59,60]. Interestingly, similar membrane vulnerabilities and increased antibiotic susceptibility have been reported in colistin-resistant *Acinetobacter baumannii* [61,62], indicating that such LPS-mediated evolutionary trade-offs may be widespread among gram-negative bacteria.

The second pathway of low Mg$^{2+}$-dependent colistin resistance relies on the synergy between PhoPQ activation and mutations in *PA4824* and *lpxA* (observed in trajectory III, *e.g.,* P8). Unlike the *htrB2*-dependent trajectory, this route does not cause major membrane defects or collateral sensitivity. While PhoPQ-dependent lipid A modification is a known resistance mechanism, the role of LpxA in colistin resistance has not been previously described. LpxA encodes an essential acyltransferase that initiates the first step in the biosynthesis of lipid A [48], making loss-of-function mutations inviable. Indeed, how the *lpxA* mutation observed in our experiment contributes to colistin resistance is unclear. It might alter enzyme activity, impacting lipid A abundance rather than composition, which our biochemical analysis cannot detect, or have lipid A-independent functions that promote resistance. Interestingly, although the early background of this trajectory had reduced fitness, the endpoint clone didn't, indicating that other late-occurring mutations might be selected, in part, because they reverse the early fitness cost.

The high frequency of *htrB2* mutations across replicate populations (6 out of 8) is unexpected, given the associated membrane defects and fitness costs. However, this pattern may reflect a larger mutational target for null mutations in

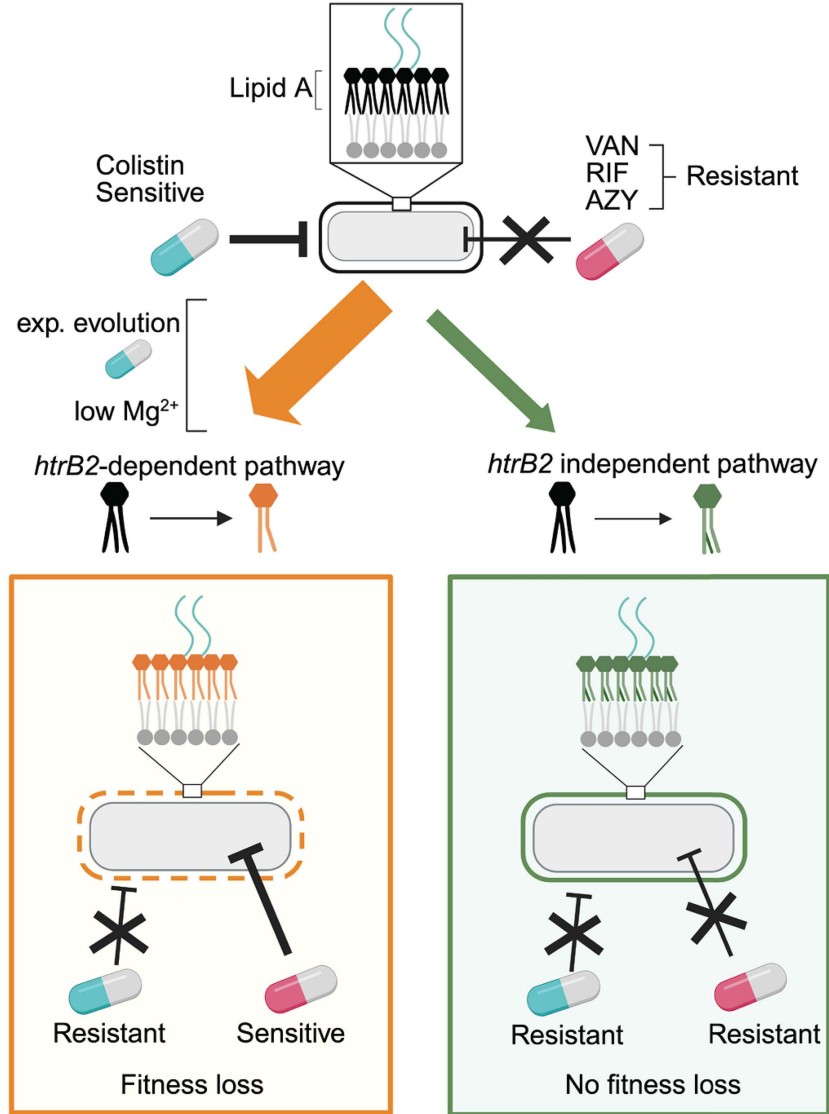

**Fig 8. Low Mg²⁺ drives two distinct evolutionary pathways to colistin resistance.** WT *Pseudomonas aeruginosa* cells (top) are sensitive to colistin, which targets the bacterial outer membrane, but resistant to other antibiotics targeting intracellular processes, including vancomycin (VAN), rifampicin (RIF), and azithromycin (AZY). After passaging in low Mg²⁺ media with gradually increasing colistin concentrations, *P. aeruginosa* evolves colistin resistance via two distinct pathways. One pathway (left) relies on *htrB2* loss-of-function mutations, leading to increased membrane permeability, collateral sensitivity to other antibiotics, and reduced fitness. The other pathway (right) does not depend on *htrB2* mutation and does not exhibit the same trade-offs. This figure is created with Biorender.com, Hsieh, P. (2026) https://BioRender.com/yyss3yk.

*htrB2* than for mutations in essential lipid A biosynthesis genes such as *lpxA*. Among the six *htrB2*-dependent populations, we identified three distinct mutations: R170W (P1 and P5), R294H (P2 and P3), and a 35-bp deletion that leads to the loss of nucleotides 517–551 (P7). In contrast, the two *lpxA*-mutant populations (P4 and P8) in trajectory III share the same G170C mutation. These findings suggest that *htrB2*-mediated resistance may arise more readily through multiple available mutational paths to ablate function.

In addition to *htrB2* and *lpxA*, one of the most intriguing genes identified in our evolution experiment is *PA4824*, a putative Mg²⁺ transporter [37]. All replicate populations incurred either the P224L or P244A mutation in *PA4,824.* In two of

three evolutionary lineages, reverting the P224L mutation to WT reduced colistin resistance. Yet, this mutation alone does not impair intracellular $Mg^{2+}$ levels, unlike the complete loss of *PA4824* (S18 Fig), suggesting that resistance is promoted independently of $Mg^{2+}$ uptake. Given that *PA4824* encodes a transmembrane protein, we suspect this mutation may stabilize the compromised outer membrane caused by other resistance-associated mutations. Future studies could investigate how the *PA4824* mutation, either alone or in combination with other mutations, affects the bacterial outer membrane to enhance colistin resistance.

We discovered several novel lipid A structures that contributed to enhanced colistin resistance in *P. aeruginosa* by reducing its binding to the membrane. Lipid A modifications that reduce colistin binding are a well-known means by which gram-negative bacteria can acquire resistance to this antibiotic. Several mechanisms have been described, including PmrAB-dependent addition of phosphoethanolamine or galactosamine to lipid A [63–65] or complete loss of LPS [62] in *Acinetobacter baumannii*, or presence of plasmid-borne phosphoethanolamine transferase in *P. aeruginosa* [66] and *Escherichia coli* [67]. In addition to PhoPQ-dependent modifications, we identified loss-of-function mutations in *htrB2* and *lpxO2*, which alter acyl chain number and hydroxylation status of lipid A and are necessary for increased colistin resistance in low $Mg^{2+}$. Interestingly, identical *htrB2* and *lpxO2* mutations have been identified in a few clinical *P. aeruginosa* isolates [68–70], while other *lpxO2* loss-of-function variants have been observed in several *P. aeruginosa* isolates obtained from patients with cystic fibrosis [41,71]. Though not previously linked to colistin resistance, we propose that these otherwise deleterious mutations may have been selected in $Mg^{2+}$-poor environments by reducing colistin binding to membranes. Such unexpected lipid A modifications could be markers of low $Mg^{2+}$-dependent colistin resistance in clinical *P. aeruginosa* isolates.

The evolution of colistin resistance has been shown to be a multi-step process, in which early mutations in *pmrB* and *phoQ* potentiate resistance, and subsequent epistasis with other mutations drives high-level resistance [72]. Our work aligns with this framework and further highlights the distinct roles of PhoPQ and PmrAB systems in shaping the evolutionary paths to colistin resistance. Although both two-component systems are activated upon $Mg^{2+}$ depletion to modify lipid A [24,29], we showed that selection for colistin resistance under chronic $Mg^{2+}$ depletion drives PhoPQ activation—but not PmrAB activation—to synergize with several lipid A-altering mutations and enhance colistin resistance during evolution. Even though these two regulatory systems co-regulate several genes, we hypothesize that specific PhoP-regulated genes [49] that PmrAB does not activate engage in key epistatic interactions with the mutations we have identified.

Our thorough genetic and biochemical analyses of colistin-resistant *P. aeruginosa* have important therapeutic implications. First, the evolutionary trade-offs suggest that pairing colistin with other intracellular-targeting antibiotics might more effectively overcome antibiotic resistance. Second, our findings reveal the importance of PhoPQ in maintaining cell morphology under low $Mg^{2+}$ conditions, suggesting a strategy in which PhoQ-specific kinase inhibitors could be used alongside colistin to prevent low $Mg^{2+}$-dependent colistin resistance [73].

Together, our findings underscore the critical role of $Mg^{2+}$ in shaping the evolution of antibiotic resistance. We demonstrate that bacterial evolution in low $Mg^{2+}$ environments, such as during fungal-bacterial interactions [37], and within polymicrobial biofilms [38,74], can lead to unconventional evolutionary pathways to resistance against colistin (and possibly other antibiotics). These adaptations often, but not always, incur trade-offs that can be exploited for therapeutic gain. Our study expands the mechanistic understanding of colistin resistance and its vulnerabilities, and highlights the ecological relevance of metal ions in driving the evolution of antibiotic resistance. We previously showed that $Mg^{2+}$ sequestration by fungi extends beyond *P. aeruginosa* to other gram-negative bacteria [37]. Our current study lays a foundation for developing evolution-guided strategies to combat multidrug-resistant *P. aeruginosa* and other gram-negative bacteria. More broadly, since many other antibiotic mechanisms are similarly dependent on metal ions [75–77], our work suggests that nutritional competition for metal ions may alter initial antibiotic resistance and potentiate new evolutionary pathways of antibiotic resistance.

## Materials and methods

### Fungal and bacterial strains used in this study

Bacterial and fungal strains used in this study are listed in S3 Table. Bacterial strains in this study were derived from *P. aeruginosa* PAO1. Unless otherwise specified, all experiments were performed in Brain Heart Infusion Broth (BHI, Sigma-Aldrich), buffered with 10% MOPS and 2% glucose to pH 7.0, and then filter-sterilized. All strains were grown at 30 or 37 °C. For antibiotics used in this study, 100 µg/mL gentamicin was used to select against *P. aeruginosa*. Fifty µg/mL nystatin was used to select against *C. albicans*. Colistin (Sigma-Aldrich) was prepared as a 10 mg/mL stock solution.

### Co-culture colistin survival assay

All experiments began with bacterial or fungal cultures that were grown overnight. Starter cultures were diluted either 1:100 (for bacterial cultures) or 1:50 (for fungal cultures) in BHI and cultured for ~4–5 hours to reach the logarithmic phase. Refreshed bacterial and fungal strains were added to BHI to achieve final bacterial cell densities of $2.5 \times 10^4$ cells/mL and fungal cell densities of $5 \times 10^5$ cells/mL for co-culture experiments. Equal numbers of bacterial or fungal cells were added separately to fresh BHI media as monoculture controls. These cultures were incubated for 18 h at 37 °C with shaking. To equalize bacterial cell numbers in these two conditions before colistin treatment, the cell density of monoculture samples was adjusted to an optical density of 0.3 at 600 nm ($OD_{600}$) to match the bacterial cell density after 18 h of growth in co-culture. Both equilibrated monoculture and co-culture samples were split into two 1 mL culture tubes. 192 µg/ml of colistin was added to one of the tubes. Cells in both tubes were incubated at 37 °C for 1.5 hours, and the bacterial viability was determined by enumerating colony-forming units (CFU) in each condition. Bacterial survival was calculated as the ratio of CFU after colistin treatment relative to untreated controls.

### Antibiotic MIC assays

MICs were determined using a standard serial broth dilution method. Bacterial cells were cultured in BHI media overnight at 37 °C with shaking and diluted 1:100 into 2 mL of either BHI media (high $Mg^{2+}$ media) or *C. albicans*-spent media (low $Mg^{2+}$ media) for 5 hours to determine MIC in high $Mg^{2+}$ and low $Mg^{2+}$ conditions, respectively. Next, 3 µL of the log-phase culture was inoculated into each well with 200 µL of the same media and titrated antibiotics (colistin from 0 to 768 µg/mL, azithromycin from 0 to 128 µg/mL, vancomycin from 0 to 1,024 µg/mL, and rifampicin from 0 to 256 µg/mL). The plate was incubated at 37 °C for 24 h, after which $OD_{600}$ was measured using a microplate reader. The MIC of each strain was determined as the lowest concentration of antibiotics at which $OD_{600}$ was half of the maximum $OD_{600}$ observed in the absence of colistin. Three technical replicates were used to measure the MIC of each strain. Additionally, to more accurately determine the maximal colistin MICs of our endpoint strains, we expanded the colistin concentration using finer fold increases (1.5×, 2×, 2.5×, 3×, 3.5×, and 4×) from 192 to 768 µg/mL.

### Mutation reconstruction in WT PAO1

We used an allelic exchange method to reconstruct mutations in WT PAO1 or revert evolved mutations to a WT allele in endpoint clones [78]. Briefly, a gene fragment with or without mutation was incorporated into the pEXG2 plasmid and transformed into *E. coli* S17. The S17 donor was subsequently mixed with *P. aeruginosa* recipients on an LB agar plate at a 5:1 ratio of donor to recipient cells, and the cell mixture was incubated at 30 °C overnight. The cell mixture was plated on LB agar containing 100 µg/mL gentamicin to select for cells carrying the deletion plasmid integrated into the *P. aeruginosa* genome. Counterselection was performed with LB agar plates containing sucrose. Gene deletions were confirmed by Sanger sequencing of PCR products (GeneWiz, Azenta). Primers and plasmids used for strain constructions are listed in S4 and S5 Tables.

## Whole genome sequencing of evolutionary intermediates of evolved populations

Populations were frozen at two-week intervals and at the endpoint of the experimental evolution [37]. We revived each population from a freezer stock and grew them for 24 h. For co-culture evolved populations, we treated them with nystatin to remove fungal cells. A 3 ml bacterial culture was collected for DNA extraction using the DNeasy Blood & Tissue kit (Qiagen). Sequencing libraries were prepared and sequenced commercially using Illumina technology by SeqCenter (https://www.seqcenter.com/). Variants were called using the breseq software v0.37.1, with the *P. aeruginosa* PAO1 genome (GCF_00006765.1) as the reference genome. The average genome coverage was 98.8 fold (±0.04). Mutations were manually curated by identifying unique variants in each evolved population compared to the ancestor. Whole-genome sequencing data of evolved populations are available in NCBI Bioproject PRJNA1251133.

## Mass spectrometry analysis of lipid A

Lipid A structural analysis was performed using FLAT, as previously described [79]. Bacterial cultures were grown to the mid-log phase and pelleted. The pellet was scraped directly onto a steel MALDI target plate in duplicate with a pipette tip. FLAT extraction buffer (1 μL; 0.2 M anhydrous citric acid, 0.1 M trisodium citrate dihydrate, pH 4.5) was pipetted over bacterial spots. The FLAT target plate was incubated at 100 °C in a humidified heat block for 30 min. Bacterial spots were washed with endotoxin-free water Gibco (Grand Island, NY, USA) and air-dried, followed by the addition of norharmane matrix (10 mg/mL in 1:2 MeOH: $CHCl_3$—Sigma Aldrich, St. Louis, MO, USA). Mass spectra were collected in negative-ion mode using a Bruker timsTOF Flex. Agilent ESI Tune Mix was used as an external calibrant. Mass spectra were processed using DataAnalysis v3.4 software. Ions for tandem mass spectrometry were identified in DataAnalysis and fragmented in the gas phase between 70 and 90 eV. Each MS analysis was performed with at least two independent biological replicates. Mass spectra were analyzed using mMass v5.5.0. S2 Table describes all the *m/z* values determined in our experiments.

## Quantification of gene expression

Log-phase cell cultures in BHI and *in C. albicans*-spent BHI were collected for RNA extraction using the RNeasy Mini Kit (Qiagen) according to the manufacturer's protocol. RNA was converted to cDNA using qScript cDNA synthesis kit (QuantaBio). The expression levels of genes of interest were determined by quantitative PCR using PowerTrack SYBR PCR mix (Thermo Fisher) and gene-specific primers (S3 Table)., relative to a housekeeping gene, *rplU*.

## Outer membrane permeability assay

We used a previously established NPN uptake assay [18,80]. Bacterial cells were grown to log phase in BHI or *C. albicans*-spent BHI. Cells were diluted to an $OD_{600}$ of 0.5 in 1 mL 5 mM HEPES, pH 7.2. 150 μl bacterial suspension was added to wells of a black microtiter plate with clear-bottomed wells. The fluorescent probe N-phenyl-1-naphthylamine (NPN, Sigma Aldrich) was added to a final concentration of 10 μM. Fluorescence was measured immediately in a Cytation 5 plate reader using an excitation wavelength of 535 nm and an emission wavelength of 405 nm. Fluorescence measurements were obtained every 1 min for 15 min, and the degree of outer membrane permeability, referred to as the NPN uptake score, was calculated using the following equation:

$$(\textit{Fluorescence of sample with NPN} - \textit{Fluorescence of sample without NPN})/$$
$$(\textit{Fluorescence of HEPES buffer with NPN} - \textit{Fluorescence of HEPES buffer without NPN}).$$

## Scanning electron microscopy experiments

1 mL of log-phase cells in BHI was collected and fixed with formaldehyde. Duplicates of 50 μL of each sample were applied in a pool on poly-l-lysine-coated coverslips for 30 min, rinsed with 0.1M sodium cacodylate buffer, and

treated with 1% osmium tetroxide for 1 hour. The coverslips were then rinsed with cacodylate buffer, dehydrated through a graded series of alcohols, infiltrated with HMDS (Electron Microscopy Sciences, Hatfield, PA), and air-dried. Coverslips were mounted on stubs and sputter-coated with gold/palladium (Denton Desk IV, Denton Vacuum, Moorestown, NJ). Samples were imaged on a JSM-6610LV SEM at 15 kV and a working distance of 12 mm (JEOL, Tokyo, Japan).

## Polymyxin B binding assay

Bacterial strains were cultured in BHI media overnight at 37 °C with shaking. Cultures were then diluted at 1:100 in BHI (high $Mg^{2+}$) and *C. albicans*-spent BHI (low $Mg^{2+}$) for 5 hours to reach log phase. 1 mL of log-phase cells was collected, washed, and resuspended in 1 mL 0.9% NaCl to reach 0.5 $OD_{600}$. 1 mL of cell suspension was then mixed with 3 µg/mL dansyl-polymyxin B (Sigma-Aldrich) for 30 min in the dark. After incubation, 150 µL of each suspension was transferred into wells of a black 96-well microtiter plate to measure fluorescence (excitation at 340 nm and emission at 485 nm). The dansyl fluorophore shows low fluorescence in aqueous solution but exhibits enhanced fluorescence upon binding to bacterial membrane LPS. To calculate polymyxin B binding to cells, we use the following formula to measure changes in fluorescence intensity:

$$fluorescence\ change\ (\%) = \frac{F - F0}{F0} \times 100,$$

where $F$ = fluorescence intensity of sample with dansyl-polymyxin B, and $F0$ = fluorescence intensity of sample with dansyl-polymyxin B. The relative fluorescence intensity of each sample was normalized to WT under the same condition.

## Competitive fitness assay

WT PAO1 with chromosomally integrated mCherry was used as the reference strain in fitness competition with evolved strains. The relative fitness of each evolved strain was determined either in BHI media only (high $Mg^{2+}$ condition) or in BHI media in co-culture with *C. albicans* (low $Mg^{2+}$ condition) to recapitulate the culturing conditions of experimental evolution. All bacterial and yeast strains were incubated in BHI at 37 °C to log phase. A non-fluorescent test strain was mixed with the fluorescence-labeled reference strain at a ratio of 1:1. Part of the cell mixture was used to determine the initial ratio of sample and reference strains using flow cytometry (BD FACSymphony A5 Cell Analyzer). Then, 10 µl of cell mixture was inoculated separately in 2 ml BHI alone (monoculture) or 2 ml BHI with $2 \times 10^5$ *C. albicans* cells (co-culture) and cell cultures were grown in BHI at 37 °C for 18 hours. After 18 hours, monoculture samples were diluted 100-fold to measure the ratio of sample and reference strain. For co-culture samples, a low-speed spin (500$g$ for 8 min) was applied to separate bacterial and fungal populations. Supernatants enriched with bacterial cells were identified using flow cytometry. At least 30,000 cells were collected for each sample, and the data were visualized using FlowJo 10.4.1. The reference strain was cultured separately to estimate the number of generations during the experiment. Each experiment was conducted in at least two biological and two technical replicates. To calculate the relative fitness, $w$, of each sample strain to the reference strain, we followed the formula: $w = 1 + s$,

$$where\ s\ is\ selection\ coefficient:\ s = \frac{\ln\left(\frac{sample}{reference}\right)t - \ln\left(\frac{sample}{reference}\right)0}{t},$$

where $t$ = number of generations and (sample/reference) is the ratio between a sample strain and the reference strain [81].

# Supporting information

**S1 Fig. Schematic of lipid A biosynthesis and modification pathway. (A)** Lipid A biosynthesis in gram-negative bacteria begins with the synthesis of lipid IV$_A$, which consists of phosphorylated glucosamine sugars, each of which serves as the backbone for an N-linked 3′OH-C12 and O-linked 3′OH-C10 acyl chain in *P. aeruginosa*. First, LpxA catalyzes the transfer of an acyl group to UDP-N-acetylglucosamine (UDP-GlcNAc), forming UDP-3-O-acyl-GlcNAc. Subsequently, the HtrB1/LpxO2 and HtrB2/LpxO1 acyltransferases and hydroxylases add secondary O-linked C12 acyl chains and 2′ hydroxyl groups at the 3′OH residue of the N-linked acyl chains; such regulation is very conserved in gram-negative bacteria. Lipid biosynthesis/modification mutations observed in our evolution experiment are bolded (LpxA, HtrB2, and LpxO2). In certain cases, PagL removes the acyl chain from the 3-position of lipid A. **(B)** Low Mg$^{2+}$-responsive PhoPQ pathway upregulates enzymes that modify lipid A structures [28,29]. These include *pagP*, which encodes an acyl transferase that adds a C16 fatty acid, and the *arn* operon (*arnBCADTEF*), which attaches 2-amino-2-hydroxy-L-arabinose (aminoarabinose) to the phosphate groups on the glucosamine backbone of lipid A.
(EPS)

**S2 Fig. Triple-mutation-reconstructed strains show significantly increased survival to a range of low concentrations of colistin in co-culture.** *P. aeruginosa* survival to low concentrations of colistin in co-culture was measured using the colistin survival assay. Survival of triple mutants to 6, 12, and 24 μg/ml colistin is shown in green, blue, and magenta, respectively. P2-derived triple mutant contains *htrB2*, *PA4824*, and the *oprH*/*phoPQ* promoter mutations. P5-derived triple mutant contains *htrB2*, *PA4824*, and *lpxO2* mutations. P8-derived triple mutant contains *lpxA*, *PA4824*, and the *oprH*/*phoPQ* promoter mutations. All three triple mutants had higher survival at these colistin concentrations than WT. Mean ± std of 4 biological replicates is shown. The survival of WT PAO1 in three conditions is used as the reference for statistical analysis (\*\**p* < 0.01, \**p* < 0.05, one-way ANOVA test with Dunn's multiple comparison correction). The underlying data of this figure can be found in S8 Data.
(EPS)

**S3 Fig. Mutation reversion strains show reduced colistin MIC in low Mg$^{2}$+ media.** Colistin resistance in endpoint clones, single-mutation revertant strains, and gene-deletion mutants was measured using colistin MIC assays. Colistin resistance of strains derived from P2, P5, and P8 is shown in D, E, and F. Mean ± std of 3 biological replicates is shown. The MIC of each endpoint clone is used as the reference for statistical analysis (\* *p* < 0.05; Kruskal–Wallis test with Dunn's multiple-comparison correction). The fold change in MIC relative to the endpoint clones is shown above the raw data. The underlying data of this S3A–S3C Fig can be found in S9 Data.
(EPS)

**S4 Fig. Molecular structures of lipid A in the WT PAO1 strains and endpoint clones. (A)** Molecular structure of lipid A (*m/z* = 1445.86) of WT PAO1 in high Mg$^{2+}$ media. **(B)** Molecular structure of lipid A (*m/z* = 1576.82) of WT PAO1 in low Mg$^{2+}$ media. **(C)** Molecular structure of lipid A (*m/z* = 1362.98) of P2 in low and high Mg$^{2+}$ media. **(D)** Molecular structures of lipid A (*m/z* = 1378.75, 1560.92) of P5 in low and high Mg$^{2+}$ media. **(E)** Molecular structure of lipid A (*m/z* = 1815.14) of P8 in low and high Mg$^{2+}$ media.
(EPS)

**S5 Fig. Similar mass spectra of evolved populations in high and low Mg$^{2+}$ conditions. (A–H)** Mass spectra of lipid A for P1 through P8 are shown in low (above the horizontal line) versus high Mg$^{2+}$ (below the horizontal line) conditions. All lipid A peaks present in low Mg$^{2+}$ media are also present in high Mg$^{2+}$ conditions. Variations in total peak height are due to variations in absolute peak intensity between samples. Populations 1, 5, and 7 contain tetra-acylated lipid A lacking HtrB2-mediated C12 addition, penta-acylated lipid A with HtrB2-mediated C12 addition, and all lipid A lacks LpxO1/2-mediated 2′-hydroxylation. Populations 2, 3, and 6 contain penta-acylated lipid A with PagP-mediated palmitoylation and

lack HtrB2-mediated C12 addition. Populations 4 and 8 contain hexa-acylated lipid A with PagP-mediated palmitoylation, and are singly 2′-hydroxylated, lacking LpxO1-mediated 2′-hydroxylation. See S1 Table for further details of every lipid A structure.
(EPS)

**S6 Fig. Mass spectra of triple-mutation-reconstructed strains resemble those from endpoint populations.** Mass spectra of lipid A from all triple mutants. **(A)** Lipid A MS of P2 triple reconstruction mutant. Peaks at m/z 1501.92, 1632.98, and 1764.04 were present in the P2 endpoint clone, representing penta-acylated lipid A with PagP-mediated C16 addition and no HtrB2-mediated C12 addition. **(B)** Lipid A MS of the P5 triple mutant reconstruction. All peaks present were also present in the P5 endpoint clone, with peaks at $m/z$ 1247.69, 1378.75, and 1509.81 representing tetra-acylated lipid A lacking HtrB2-mediated C12 addition. Peaks at $m/z$ 1429.86 and 1560.92 represent penta-acylated lipid A containing HtrB2-mediated C12 addition, but no LpxO1 or LpxO2-mediated 2′-hydroxylation. **(C)** Lipid A MS of P8 triple mutant reconstruction. Peaks at 1684.08, 1815.14, and 1946.20 were all present in the P8 endpoint clone, representing hexa-acylated lipid A with a single 2′-hydroxylation mediated by LpxO2, and PagP-mediated palmitoylation. Peaks at 1445.86 and 1576.92 were not present in the P8 endpoint clone and represent penta-acylated lipid A without PagP-mediated palmitoylation. All peaks that are +15.99 $m/z$ from the previously mentioned peaks contain LpxO1-mediated 2-hydroxylation, which is also not present in the original P8. See S1 Table for further information on all lipid A structures present. **(D)** Principal component analysis of lipid A mass spectra of triple mutant strains (P2, P5, and P8), all the evolved endpoint strains, and the WT. Triple mutants of P2 and P5 show PC values that are almost identical to those of the corresponding endpoint strains. The triple mutant of P8 is clustered more closely with the P8 endpoint strain than with other final evolved strains.
(EPS)

**S7 Fig. Mass spectra of mutation reversion strains.** Mass spectra of lipid A from all mutation reversion strains. **(A)** MS of WT *htrB2* in P2. Peaks present indicate hexa-acylated lipid A with HtrB2 function restored. **(B)** MS of WT *PA4824* in P2. All peaks present were also found in the original P2. **(C)** WT *oprH/phoPQ* promoter in P2. Peaks present indicate tetra- and penta-acylated lipid A lacking in palmitoylation. Single aminoarabinose addition peaks are still present. **(D)** MS of WT *htrB2* in P5. Peaks present indicate penta-acylated lipid A with the restored function of htrB2. **(E)** MS of WT *PA4824* in P5. All peaks present were also found in the original P5. **(F)** WT *lpxO2* in P5. Peaks present include tetra- and penta-acylated, singly hydroxylated lipid A. **(G)** MS of WT *oprH/phoPQ* promoter in P8. Peaks present indicate penta-acylated lipid A with varying phosphorylation levels, indicating a lack of PagP-mediated palmitoylation and of L-Ara4N addition. **(H)** MS of WT *PA4824* in P8. All peaks present were also found in the original P8. See S1 Table for further details of all lipid A structures present. **(I)** Principal component analysis of mutation reversion lipid A mass spectra. Overall, mutation reversions cluster with other endpoint clones or are similar to wild-type PAO1, depending on the reversion. Clusters are labeled based on the endpoint clone they are most like. Reversion of PA4824 to wild-type clusters is very close to each endpoint clone. Reversion of *PoprH* in P2 clusters closely to P5. Reversion of *htrB2* in P2 clusters is close to P8. Reversion of *PoprH* in P8 clusters is close to wild-type PAO1. Reversion of *htrB2* and *lpxO2* in P5 cluster close to PAO1, but are still distinct.
(EPS)

**S8 Fig. *phoP, phoQ, pmrA,* and *pmrB* genes are induced when WT PAO1 grows in low Mg$^{2+}$ media.** Log-phase cultures of WT PAO1 in high- and low-Mg$^{2+}$ media were processed for RNA extraction. qPCR with gene-specific primers was used to determine the expression of *phoP, phoQ, pmrA,* and *pmrB* genes, as described in the Materials and methods. Gene expression is shown as expression relative to the internal control rplU and normalized to high Mg$^{2+}$ media. All these genes were upregulated in the low Mg$^{2+}$ media. Mean ± std of 3 biological replicates are shown (**$p < 0.01$, *$p < 0.05$, one-tailed Mann–Whitney $U$ test). The underlying data of this figure can be found in S10 Data.
(EPS)

**S9 Fig. Colistin resistance in P2, P5, and P8 requires PhoPQ activity.** Δ*pmrA*, Δ*phoP*, and Δ*phoP* Δ*pmrA* were made in the triple mutants **(A)**, endpoint clones **(B)**, and WT PAO1 **(C)**. The standard MIC assay in the low Mg$^{2+}$ media measured colistin resistance. In both evolutionary backgrounds, *phoP* deletion significantly reduced colistin MIC, but *pmrA* deletion had a mild effect. Mean ± std of 3 biological replicates is shown. The fold change in MIC relative to WT is shown above the raw data (*$p < 0.05$, One-tailed Mann–Whitney *U* test). The underlying data of S9A–S9C Fig can be found in S11 Data. (EPS)

**S10 Fig. Deletion of *phoP*, but not *oprH*, in P2 and P8 clones reduced colistin resistance.** *phoP* or *oprH* was deleted in the P2 and P8 endpoint clones to distinguish which gene is required for evolved colistin resistance. MICs measured in low Mg$^{2+}$ media showed colistin resistance. *phoP* deletion, but not *oprH* deletion, significantly reduced colistin MIC. Mean ± std of 3 biological replicates is shown. (**$p < 0.01$, One-tailed Mann–Whitney *U* test). The underlying data of this figure can be found in S12 Data. (EPS)

**S11 Fig. Mass spectra of Δ*phoP* in the endpoint strains of P2 and P8.** Δ*phoP* in P2 **(A)** and P8 **(B)** show lipid A without aminoarabinose addition and PagP-mediated acylation. **(A)** Mass spectra of Δ*phoP* in P2 yielded only one lipid A peak at 1263.68 *m*/*z*. This corresponds to a tetra-acylated lipid A structure lacking the previously present aminoarabinose additions and PagP-mediated palmytoylation in P2. **(B)** FLAT followed by MALDI-TOF MS of Δ*phoP* in P8 yielded only a single lipid A ion corresponding to the penta-acylated, singly hydroxylated 1445.89 *m*/*z*, lacking the aminoarabinose additions and PagP-mediated C16 addition that were present in the final P8 evolved clone. (EPS)

**S12 Fig. Replicates of SEM images of endpoint clones in the high and low Mg$^{2+}$ media.** In the high Mg$^{2+}$ media, P2 and P5 endpoint clones, but not P8 endpoint clones, have discernible dents or kinks in the cell membrane (white arrows). In low Mg$^{2+}$ media, P5 showed membrane deformation. All three displayed altered cell shapes and lengths compared to WT PAO1. The scale bar indicates 1 μm. (PDF)

**S13 Fig. Three endpoint clones showed higher polymyxin B MIC than the WT PAO1. (A)** Polymyxin B MIC of endpoint clones in high Mg$^{2+}$ and low Mg$^{2+}$ media. Mean ± std of 3 biological replicates is shown. The fold change in MIC relative to WT is shown above the raw data (* $p < 0.05$, Kruskal–Wallis test with Dunn's multiple comparison correction). **(B)** The lack of PhoPQ and PmrAB activity increased polymyxin B (3 μg/mL) binding to the WT PAO1 cells. Dansyl-polymyxin B was used to measure polymyxin B binding to bacteria. Mean ± std of 3 biological replicates is shown (** $p < 0.01$, one-way ANOVA test with Dunn's multiple comparison correction). The underlying data of S13A and S13B Fig can be found in S13 Data. (EPS)

**S14 Fig. Fitness cost of triple mutants in high and low Mg$^{2+}$ conditions.** A competitive fitness assay was used to assess the fitness of three triple mutants relative to WT in high Mg$^{2+}$ (blue) and low Mg$^{2+}$ conditions (orange). Triple mutants of P2 and P5 had reduced fitness in both conditions, whereas the P8 triple mutant showed a fitness cost only in high Mg$^{2+}$ media. Mean ± std of 4 biological replicates is shown. (**$p < 0.01$, one-way ANOVA test with Dunn's multiple comparison correction). The underlying data of this figure can be found in S14 Data. (EPS)

**S15 Fig. P2 and P5 show lower colistin resistance in high Mg$^{2+}$ media than in low Mg2$^+$ media.** Colistin resistance of triple mutants **(A)** and endpoint clones **(B)** of P2 and P5 was measured by colistin MIC assay in high and low Mg$^{2+}$ media. In both lineages, cells showed significantly lower MIC in the high Mg$^{2+}$ media compared to the low Mg$^{2+}$ media. Mean ± std

of 3 biological replicates is shown. The fold change in MIC relative to WT is shown below the raw data (**$p < 0.01$, Mann–Whitney $U$ test). The underlying data of S15A–S15C Fig can be found in S15 Data.
(EPS)

**S16 Fig. Correlation between fitness and antibiotic susceptibility in high and low Mg$^{2+}$ conditions.** The correlation between the fitness and antibiotic MICs of P2, P5, P8 endpoint clones, and WT PAO1 is shown under high Mg$^{2+}$ **(A, C, and E)** and low Mg$^{2+}$ **(B, D, and F)** conditions. The Pearson correlation coefficient is indicated in each panel. The underlying data of S16A–S16F Fig can be found in S16 Data.
(EPS)

**S17 Fig. Monoculture evolved populations don't have the trade-offs between colistin resistance and susceptibility to other antibiotics.** Endpoint clones of monoculture-evolved populations (M2, M5, and M8, evolved for higher colistin resistance in Mg$^{2+}$-replete conditions), derived from the same ancestral clone as P2, P5, and P8, were assayed for MICs to colistin **(A)**, vancomycin **(B)**, azithromycin **(C)**, and rifampicin **(D)**. The fold change in MIC relative to WT is shown above the raw data. Their membrane permeability and relative fitness are shown in **(E)** and **(F)**, respectively. In contrast to P2, P5, and P8, M2, M5, and M8 don't show increased antibiotic susceptibility and fitness cost associated with more severe membrane defects than WT PAO1. The underlying data of S17A–S17F Fig can be found in S17 Data.
(EPS)

**S18 Fig. *PA4824* mutation has no clear effect on altering intracellular Mg$^{2+}$ levels.** A Mg$^{2+}$ genetic reporter assay published previously [37] was used to measure the relative intracellular Mg$^{2+}$ levels of WT, Δ*PA4824*, and *PA4824* (P224L) single mutant in high Mg$^{2+}$ and low Mg$^{2+}$ conditions. *PA4824* mutation reconstructed strain had subtle effects on Mg$^{2+}$ levels, indicating this mutation might not impair Mg$^{2+}$ transport function significantly. Mean ± std of 3 biological replicates is shown (**$p < 0.01$, *$p < 0.05$, one-way ANOVA test with Dunn's multiple comparison correction). The underlying data of this figure can be found in S18 Data.
(EPS)

**S1 Table. All the mutations of P2, P5, and P8 endpoint clones in this study.**
(DOCX)

**S2 Table. Peak table of all major and minor lipid A species present in MS analysis.** Every *m/z* peak representative of lipid A structure found in FLAT followed by MALDI-TOF MS. C3′ and C2′ refer to the third and second carbons on the glucosamine on the left side, while C3 and C2 refer to the third and second carbons of the right glucosamine, respectively. Phosphorylation status and aminoarabinose status are not site-specific.
(DOCX)

**S3 Table. Strains used in this study.**
(DOCX)

**S4 Table. Primers used in this study.**
(DOCX)

**S5 Table. Plasmids used in this study.**
(DOCX)

**S1 Data. The data underlying Figs 1D–1F.**
(XLSX)

**S2 Data. The data underlying Figs 2A–2F.**
(XLSX)

**S3 Data. The data underlying** Fig 3G.
(XLSX)

**S4 Data. The data underlying** Figs 4A **and** 4B.
(XLSX)

**S5 Data. The data underlying** Figs 5B–5E.
(XLSX)

**S6 Data. The data underlying** Figs 6A **and** 6B.
(XLSX)

**S7 Data. The data underlying** Figs 7B–7D.
(XLSX)

**S8 Data. The data underlying** S2 Fig.
(XLSX)

**S9 Data. The data underlying** S3 Fig.
(XLSX)

**S10 Data. The data underlying** S8 Fig.
(XLSX)

**S11 Data. The data underlying** S9 Fig.
(XLSX)

**S12 Data. The data underlying** S10 Fig.
(XLSX)

**S13 Data. The data underlying** S13 Fig.
(XLSX)

**S14 Data. The data underlying** S14 Fig.
(XLSX)

**S15 Data. The data underlying** S15 Fig.
(XLSX)

**S16 Data. The data underlying** S16 Fig.
(XLSX)

**S17 Data. The data underlying** S17 Fig.
(XLSX)

**S18 Data. The data underlying** S18 Fig.
(XLSX)

## Acknowledgments

We thank the Ernst, Dandekar, and Malik lab members for their valuable discussions on this project. We thank Pete Greenberg, Carrie Harwood, Nina Salama, and Andrew Murray for their comments on the manuscript. We especially thank Nina Salama for her suggestion to investigate lipid A modifications and Steve MacFarlane and the Fred Hutch Electron Microscopy & CryoEM Core for supporting scanning electron microscopy experiments.

## Author contributions

**Data curation:** Yu-Ying Phoebe Hsieh.

**Formal analysis:** Yu-Ying Phoebe Hsieh.

**Funding acquisition:** Yu-Ying Phoebe Hsieh, Robert K Ernst, Ajai A. Dandekar, Harmit S Malik.

**Investigation:** Yu-Ying Phoebe Hsieh, Ian P. O'Keefe, Zeqi Wang, Wanting Sun, Hyojik Yang, Linda M Vu, Nicole E. Smalley.

**Methodology:** Yu-Ying Phoebe Hsieh, Ian P. O'Keefe, Zeqi Wang, Wanting Sun.

**Project administration:** Yu-Ying Phoebe Hsieh.

**Supervision:** Yu-Ying Phoebe Hsieh, Robert K Ernst, Ajai A. Dandekar, Harmit S Malik.

**Visualization:** Yu-Ying Phoebe Hsieh, Ian P. O'Keefe, Zeqi Wang, Wanting Sun, Robert K Ernst, Ajai A. Dandekar, Harmit S Malik.

**Writing – original draft:** Yu-Ying Phoebe Hsieh, Ian P. O'Keefe.

**Writing – review & editing:** Yu-Ying Phoebe Hsieh, Ian P. O'Keefe, Zeqi Wang, Wanting Sun, Hyojik Yang, Linda M Vu, Robert K Ernst, Ajai A. Dandekar, Harmit S Malik.

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
