## [Editor Report · Decision Letter 0]

11 Sep 2025

Dear Phoebe,

Thank you for submitting your Review Commons manuscript entitled "Magnesium depletion unleashes two unusual modes of colistin resistance with different fitness costs" for consideration as a Research Article by PLOS Biology.

Your manuscript, and the reviews and your Revision Plan, have now been evaluated by the PLOS Biology editorial staff, as well as by an academic editor with relevant expertise, and I'm writing to let you know that we would like to consider your manuscript further.

However, before we can send a Major Revision decision inviting you to revise the manuscript, we need you to complete your submission by providing the metadata that is required for full assessment. To this end, please login to Editorial Manager where you will find the paper in the 'Submissions Needing Revisions' folder on your homepage. Please click 'Revise Submission' from the Action Links and complete all additional questions in the submission questionnaire.

Once your full submission is complete, your paper will undergo a series of checks. After your manuscript has passed the checks I'll send the Major Revision decision. To provide the metadata for your submission, please Login to Editorial Manager (https://www.editorialmanager.com/pbiology) within two working days, i.e. by Sep 15 2025 11:59PM.

Kind regards,

Roli

Roland Roberts, PhD

Senior Editor

PLOS Biology

rroberts@plos.org

---

## [Editor Report · Decision Letter 1]

12 Sep 2025

Dear Phoebe,

Thank you for your patience while we assessed your manuscript "Magnesium depletion unleashes two unusual modes of colistin resistance with different fitness costs" and read the Review Commons reviews and your revision plan. These have now been evaluated by the PLOS Biology editors and an Academic Editor with relevant expertise

In light of the Review Commons reviews, we would like to invite you to revise the work to thoroughly address the reviewers' reports, broadly along the lines of your revision plan.

IMPORTANT: In case it's helpful, the Academic Editor said the following:

"I think this is an interesting manuscript. The topic is certainly timely and relevant. The work is technically strong, and the manuscript is well-written. One could probably argue that large parts are better suited to a specialized audience, and that some aspects seem anecdotal. However, I feel that the work is strong enough overall to be considered for PLOS Biology. Revision Plan: Some of reviewer #1's requests (e.g., additional bacterial species and in vivo models) are excessive, and I agree that the authors do not need to perform these experiments. Furthermore, some of reviewer #3's more serious concerns seem to be misunderstandings. In any case, I am optimistic that the reviewers will be satisfied with the planned revisions."

Given the extent of revision needed, we cannot make a decision about publication until we have seen the revised manuscript and your response to the reviewers' comments. Your revised manuscript is likely to be sent for further evaluation by all or a subset of the reviewers.

**IMPORTANT - SUBMITTING YOUR REVISION**

*Re-submission Checklist*

*Published Peer Review*

*PLOS Data Policy*

*Blot and Gel Data Policy*

Sincerely,

Roli

Roland Roberts, PhD

Senior Editor

PLOS Biology

rroberts@plos.org

REVIEWS: see REVIEW COMMONS.

---

## [Decision Letter · Decision Letter 2]

26 Jan 2026

Dear Phoebe,

Thank you for your patience while we considered your revised manuscript "Magnesium depletion unleashes two unusual modes of colistin resistance with different fitness costs" for publication as a Research Article at PLOS Biology. This revised version of your manuscript has been evaluated by the PLOS Biology editors, the Academic Editor, and the original Review Commons reviewers.

Based on the reviews and our Academic Editor's assessment of your revision, we are likely to accept this manuscript for publication, provided you satisfactorily address the remaining points raised by the reviewers, and the following data and other policy-related requests.

IMPORTANT - please attend to the following:

a) Please include the species names in the Title, i.e. "Magnesium depletion by Candida albicans unleashes two unusual modes of colistin resistance in Pseudomonas aeruginosa with different fitness costs"

b) Please address the remaining requests from reviewers #2 and #3.

c) Please address my Data Policy requests below; specifically, we need you to supply the numerical values underlying Figs 1DEF, 2ABCDEF, 3ABCDEG, 4ABCD, 5BCDE, 6AB, 7ABCD, S2, S3ABC, S5ABCDEFGH, S6ABCD, S7ABCDEFGHI, S8, S9ABC, S10, S11AB, S13AB, S14, S15ABC, S16ABCDEF, S17ABCDEF, S18, either as a supplementary data file or as a permanent DOI’d deposition.

d) Please cite the location of the data clearly in all relevant main and supplementary Figure legends, e.g. “The data underlying this Figure can be found in S1 Data” or “The data underlying this Figure can be found in https://zenodo.org/records/XXXXXXXX

e) Please make any custom code available, either as a supplementary file or as part of your data deposition.

f) Please include the URLs of your funders in the Financial Disclosure statement.

We expect to receive your revised manuscript within two weeks.

*Published Peer Review History*

*Press*

Sincerely,

Roli

Roland Roberts, PhD

Senior Editor

rroberts@plos.org

PLOS Biology

DATA POLICY:

Regardless of the method selected, please ensure that you provide the individual numerical values that underlie the summary data displayed in the following figure panels as they are essential for readers to assess your analysis and to reproduce it: Figs 1DEF, 2ABCDEF, 3ABCDEG, 4ABCD, 5BCDE, 6AB, 7ABCD, S2, S3ABC, S5ABCDEFGH, S6ABCD, S7ABCDEFGHI, S8, S9ABC, S10, S11AB, S13AB, S14, S15ABC, S16ABCDEF, S17ABCDEF, S18. NOTE: the numerical data provided should include all replicates AND the way in which the plotted mean and errors were derived (it should not present only the mean/average values).

CODE POLICY

Per journal policy, if you have generated any custom code during the course of this investigation, please make it available without restrictions. Please ensure that the code is sufficiently well documented and reusable, and that your Data Statement in the Editorial Manager submission system accurately describes where your code can be found. More information on our Code Policy, what and how to share can be found here: https://journals.plos.org/plosbiology/s/code-availability

DATA NOT SHOWN?

REVIEWERS' COMMENTS:

Reviewer #1:

[identifies himself as Samir Giri]

This revised version of the manuscript shows a significant improvement in clarity. The authors have provided thoughtful and satisfactory responses to all my previous comments. I have no further remarks and would like to thank the authors for their excellent work on this study.

Reviewer #2:

I reviewed this manuscript a while a go through the Review Commons platform. Now, I see that it is under consideration with PLOS Biology. I was already very positive during the initial round of review and I'm still positive. The authors have adequately addressed my comments. This is strong and very detailed work on an important topic.

I have only one minor comment. The sentence on lines 427+428 starting with "As expected …" is awkward. First, the end-point resistance of P2, P5, P8 was already known from the previous work by the same authors. Moreover, the reference to Fig. 5C makes little sense as this figure shows membrane permeability and not resistance patterns. Please check these inconsistencies.

Reviewer #3:

This is a revised version of a previously assessed manuscript on how Mg levels can alter the evolution of colistin resistance in Pseudomonas aeruginosa. The authors have largely addressed all the previous concerns, and I have only a few minor comments:

1. It would be useful if the authors could list the specific mutations that are present in the chosen endpoint clones (shown in e.g. Fig.2 D-F). This will make it easier to interpret the reversion mutants.

2. Some of the figure call-outs appear to be wrong. E.g.

Lines 190-194 refer to Figures 2D-F, and mention survival, but both Figures 2D-F and Fig S3 show MICs, so it is not clear where the reversion data is shown.

Line 316: It should be Figure S10 instead of S9A-B.

3. On lines 454-455, the authors state "However, the same mutations incur high fitness costs under high Mg2+ conditions, explaining why they are unlikely to occur in Mg2+-replete conditions."

However, the authors show in Fig. 6A that the fitness costs are seen in both high and low Mg conditions, and that the likely reason these mutations are not seen in Mg-replete conditions is that they don't confer resistance under those conditions (as the authors mention and show on lines 391-393 and in Fig S15). Thus, that sentence needs to be revised.

4. On lines 264-266 the authors conclude that the htrB2 evolved mutations were partial loss-of-function alleles. But in the Discussion, on line 459, they are called "null mutations in htrB2" - this should be fixed.

5. Line 486: At what position in the gene/protein, is the 35bp deletion?

---

## [Editor Report · Decision Letter 3]

11 Feb 2026

Dear Phoebe,

Thank you for the submission of your revised Research Article "Magnesium depletion by Candida albicans unleashes two unusual modes of colistin resistance in Pseudomonas aeruginosa with different fitness costs" for publication in PLOS Biology. On behalf of my colleagues and the Academic Editor, Tobias Bollenbach, I'm pleased to say that we can in principle accept your manuscript for publication, provided you address any remaining formatting and reporting issues. These will be detailed in an email you should receive within 2-3 business days from our colleagues in the journal operations team; no action is required from you until then. Please note that we will not be able to formally accept your manuscript and schedule it for publication until you have completed any requested changes.

Sincerely,

Roli

Senior Editor

PLOS Biology

rroberts@plos.org